# Diss-l-ECT: Dissecting Graph Data with local Euler Characteristic Transforms

## Abstract

The Euler Characteristic Transform (ECT) is an efficiently-computable geometrical-topological invariant that characterizes the *global* shape of data. In this paper, we introduce the *Local Euler Characteristic Transform* ($\ell$-ECT), a novel extension of the ECT particularly designed to enhance expressivity and interpretability in graph representation learning. Unlike traditional Graph Neural Networks (GNNs), which may lose critical local details through aggregation, the $\ell$-ECT provides a lossless representation of local neighborhoods. This approach addresses key limitations in GNNs by preserving nuanced local structures while maintaining global interpretability. Moreover, we construct a rotation-invariant metric based on $\ell$-ECTs for spatial alignment of data spaces. Our method exhibits superior performance than standard GNNs on a variety of node classification tasks, particularly in graphs with high heterophily.

## 1 Introduction

While traditional GNNs often rely on message-passing schemes that aggregate node features, they may lose crucial local information, particularly in the case of graphs with high heterophily. These methods can obscure structural nuances that are key to understanding the relationships between nodes. The Euler Characteristic Transform (ECT), a technique from Topological Data Analysis (TDA), offers a solution as it provides a representation of the given data space. The ECT captures topological features through sublevel set filtrations across various directions. Notably, it is invertible for so-called constructible sets, ensuring that the original data can be reconstructed from its transform. This invertibility, coupled with efficient computability, positions the ECT as a powerful tool for representation learning. In this paper, we extend the ECT to local neighborhoods, presenting the Local Euler Characteristic Transform ($\ell$-ECT), a method designed to preserve local structural details while retaining global interpretability. The $\ell$-ECT captures both structural and spatial information around each data point, making it particularly advantageous for graph-based data. Although the $\ell$-ECT is based on the topological concept of the Euler characteristic, it turns out to be a fingerprint of local neighborhoods around points and should therefore rather be seen as a geometrical-topological method. Our approach specifically addresses the challenge of neighborhood aggregation in featured graphs, ensuring lossless representation of local neighborhoods of nodes. As it turns out, $\ell$-ECTs maintain critical local details, and therefore offer a more nuanced representation that can be used for various downstream tasks such as node classification. Our method is particularly effective for tasks where node feature aggregation may obscure essential differences, such as in graphs with high heterophily. Additionally, the $\ell$-ECT framework's natural vector representation makes it compatible with a wide range of machine learning models, facilitating both performance and interpretability.

**As our main contributions,** we (i) construct $\ell$-ECTs in the context of embedded simplicial complexes and theoretically investigate their expressivity in the special case of featured graphs, (ii) empirically show that this expressivity positions $\ell$-ECTs as a powerful tool for interpretable node classification, often superior to standard GNNs, and (iii) introduce an efficiently computable rotation-invariant metric based on $\ell$-ECTs which facilitates spatial alignment of geometric graphs.

## 2 BACKGROUND

**Simplicial Complexes** A *simplicial complex $K$* is a mathematical structure that generalizes graphs to model higher-order relationships and interactions. While graphs represent pairwise connections between entities using nodes and edges, simplicial complexes extend this representation to higher dimensions by including simplices such as triangles (2-simplices), tetrahedra (3-simplices), and their higher-dimensional analogues. Formally, an (abstract) simplicial complex is a finite collection of (abstract) simplices such that every face of a simplex in the collection is also in the collection, and the intersection of any two simplices is either empty or a common face. An (abstract) $k$-simplex is defined as a set of $k + 1$ vertices, denoted $(v_0, v_1, \ldots, v_k)$, where the order of the vertices does not matter. The *faces* of a $k$-simplex are all subsets of its vertices and correspond to simplices of lower dimension. For example, the faces of a 2-simplex (triangle) are its three edges (1-simplices) and three vertices (0-simplices).

**Euler Characteristic** The *Euler characteristic $\chi$* is a topological invariant that provides a summary of the shape or structure of a topological space, such as a simplicial complex. It is defined as an alternating sum of the number of simplices in each dimension:

$$\chi(K) = \sum_{k=0}^{d} (-1)^k \sigma_k(K), \tag{1}$$

where $\sigma_k(K)$ denotes the number of $k$-dimensional simplices in the simplicial complex $K$, and $d$ is the dimension of $K$. The Euler characteristic can also be expressed in terms of the ranks of the homology groups of the complex:

$$\chi(K) = \sum_{n=0}^{\infty} (-1)^n \mathrm{rank}(H_n(K)), \tag{2}$$

where $H_n(K)$ is the $n$-th homology group, and $\mathrm{rank}(H_n(K))$ is its Betti number. These Betti numbers quantify the number of connected components, holes, voids, and higher-dimensional analogues in the space. As a topological invariant, the Euler characteristic is unchanged under homeomorphisms, making it a fundamental tool for distinguishing topological spaces. In machine learning, it has been used in topological data analysis to summarize complex shapes and structures in high-dimensional data.

**Graph Neural Networks and Message Passing** *Graph Neural Networks* (GNNs) are a class of neural network models designed to operate on graph-structured data. They extend traditional neural networks by incorporating the relational structure inherent to graphs, enabling learning tasks such as node classification. The core mechanism of GNNs is *message passing*, an iterative procedure that propagates information through the graph to update node representations based on their local neighborhood. Given a graph $G = (V, E)$, where $V$ is the set of nodes and $E$ is the set of edges, each node $v \in V$ is associated with a feature vector $\mathbf{x}_v$. At each layer $t$, the node embedding $\mathbf{h}_v^{(t)}$ is updated as:

$$\mathbf{h}_v^{(t+1)} = \mathrm{UPDATE}\left(\mathbf{h}_v^{(t)}, \mathrm{AGGREGATE}\left(\{\mathbf{h}_u^{(t)} : u \in \mathcal{N}(v)\}\right)\right), \tag{3}$$

where $\mathcal{N}(v)$ denotes the set of neighbors of node $v$, and AGGREGATE and UPDATE are learnable functions parameterized by the model. The *AGGREGATE* function combines information from neighboring nodes, while the *UPDATE* function refines the node embedding. Popular choices for these functions include mean, sum, and attention mechanisms. Through multiple layers of message passing, GNNs aggregate information from larger neighborhoods, capturing both local and global graph structure.

## 3 RELATED WORK

Graph Neural Networks (GNNs) have revolutionized the field of graph representation learning by enabling end-to-end learning of node embeddings through message passing (Kipf & Welling, 2016). However, traditional GNNs face theoretical limitations that are fundamental obstructions in learning expressive representations of graph data (Xu et al., 2018). Related to the latter phenomenon,

GNNs are known to suffer from issues like oversmoothing (Zhang et al., 2023; Rusch et al., 2023) and oversquashing (Di Giovanni et al., 2023). Hamilton et al. (2017) and Velickovic et al. (2018) have addressed these issues by incorporating sampling and attention mechanisms into the message-passing paradigm. However, even these advancements often show limited performance, particularly in graphs with high heterophily. Recent work in graph machine learning started incorporating additional geometric information into architectures, leading to *geometric convolutional networks* (Pei et al., 2020), as well as *geometric graph neural networks* (Joshi et al., 2023). Moreover, the analysis of the theoretical capacity or *expressivity* of an architecture remains an ongoing avenue of research (Morris et al., 2023), with some works providing additional "topology-aware" inductive biases (Horn et al., 2022; Verma et al., 2024) to improve overall GNN expressivity.

The *Euler Characteristic Transform* (ECT) has recently become a popular tool in topological data analysis (Turner et al., 2014; Ghrist et al., 2018). Röell & Rieck (2024) have expanded on these ideas, applying them to modern machine learning problems, including shape classification. In Curry et al. (2022); Marsh et al. (2024), transformation invariance properties of ECTs are studied. All aforementioned contributions build on global Euler Characteristic Transforms, local aspects, which are necessary for our approach, are not discussed therein. The use of the ECT is just one example of increasing interest in methods for dealing with higher-order information. In this context, simplicial complexes provide a richer structure than simple graphs (Yang et al., 2022; Ebli et al., 2020), with *message passing* over high-order domains gaining much attention (Bodnar et al., 2021), and a recent position paper outlining the benefits of methods being capable of working with such domains (Papamarkou et al., 2024).

Finally, there are attempts to use tools from topological data analysis in graph learning. In Hofer et al. (2020), persistent homology is used for graph classification. Zhao & Wang (2019) take a different approach to graph classification by learning a weighted kernel based on persistent homology. Zhao et al. (2020) include topological features of graph neighbourhoods into a standard GNN, again leveraging persistent homology. To the best of our knowledge, this work is the first that makes use of local variants of the Euler Characteristic Transform (ECT) for graph learning. The novelty of this work comprises both the study of theoretical properties of local variants of the ECT as well as their empirical utility, particularly for graph representation learning.

## 4 METHODS

**Euler Characteristic Transform (ECT)**  The Euler Characteristic Transform (ECT) of an (abstract) simplicial complex $X \subset \mathbb{R}^n$ is a function $\text{ECT}(X) : S^{n-1} \times \mathbb{R} \to \mathbb{Z}$, given by

$$\text{ECT}(X)(v, t) = \chi(\{x \in X | x \cdot v \leq t\}), \tag{4}$$

where $\chi$ denotes the Euler characteristic. The interpretation of $\text{ECT}(X)$ is therefore that it scans the ambient space of $X$ in every direction and records the Euler characteristic of the sublevel sets. In Ghrist et al. (2018), it is proven that $\text{ECT}(X)$ is invertible, meaning that $X$ can be recovered from $\text{ECT}(X)$, as long as $X$ is a so-called constructible set. The main focus of this work are compact simplicial complexes which are constructible, and therefore the invertibility theorem applies in our setting.

In practice, we approximate $\text{ECT}(X)$ via $\overline{\text{ECT}}(X)_{(m,l)} := \text{ECT}(X)_{|\{v_1,\ldots,v_m\} \times \{t_1,\ldots,t_l\}}$ for uniformly distributed directions $v_1, \ldots, v_m \in S^{n-1}$ and filtration steps $t_1, \ldots, t_l \in \mathbb{R}$. Since $X$ is compact, $t_1, \ldots, t_l$ can be chosen to lie in a compact interval $[a, b]$ with $t_1 = a$ and $t_l = b$, and so that the sequence $\{t_i\}_i$ forms a uniform partition of $[a, b]$. We note that this approximation is efficiently computable and has a natural representation as a vector of dimension $m \cdot l$. Regarding the choice of the magnitudes of $m, l$ we have the following result.

**Theorem 1.** *In the above setting, the worst case convergence of $\overline{\text{ECT}}(X)_{(m,l)}$ to the true quantity $\text{ECT}(X)$ is $\mathcal{O}((\frac{\log m}{m})^{1/(n-1)} \frac{1}{l})$.*

Curry et al. (2022) prove that the aforementioned approximation actually determines the true quantity, under mild conditions and for $m, l$ large enough. We notice that both translations and scalings of $X$ in the ambient space lead to a reparametrization of $\text{ECT}(X)$, and therefore $\text{ECT}(X)$ essentially (up to a parameter change) remains unaltered under these two types of transformations.

**Local ECT ($\ell$-ECT)**   Given a simplicial complex $X \subset \mathbb{R}^n$ and a vertex $x \in X$, we define the *local ECT* of x with respect to $k \geq 0$ as

$$\ell\text{-ECT}_k(x; X) := \text{ECT}(N_k(x; X)), \tag{5}$$

where $N_k(x; X)$ denotes an appropriate local neighborhood of $x$ in $X$, whose locality scale is controlled by a parameter $k$. Usually, $N_k(x; X)$ will be either the full subcomplex of $X$ which is spanned by the $k$-hop neighbors of $x$, or the full subcomplex of $X$, which is spanned by the $k$-nearest vertices of $x$. The first important special case arises when $X$ is a 0-dimensional simplicial complex, which is just a point cloud. In this case, the full subcomplex of $X$, which is spanned by the $k$-nearest vertices of $x$, $N_k(x; X)$, is simply given by the $k$-nearest neighbors of $x$. Being based on the Euler Characteristic, the construction of $\ell$-ECTs appears to be purely topological at first glance. However, in light of the invertibility theorem, we note that $\ell$-ECT$(x; X)$ can be interpreted as a fingerprint of a local neighborhood of $x$ in $X$. The upshot is that this fingerprint can be well approximated in practice, making it possible to obtain local representations of combinatorial data embedded in Euclidean space. This approximation works by sampling $v_1, \ldots, v_m \in S^{n-1}$ and $t_1, \ldots, t_l \in \mathbb{R}$, and considering $\overline{\text{ECT}}(N_k(x; X))_{(m,l)}$, instead of $\ell$-ECT$_k(x; X)$. The latter quantity is well-computable in practice, and the approximation error can be controlled by the sample sizes $m$ and $l$, as we discussed before. Again, this approximation has a natural representation as a vector of dimension $m \cdot l$, enabling us to encode local structural information of point neighborhoods in an approximate lossless way that can readily be used by machine learning algorithms for downstream tasks.

**$\ell$-ECT in representation learning**   Our formulation of $\ell$-ECTs provides a natural representation of local neighborhoods of embedded simplicial complexes. One important special case is that of featured graphs, meaning graphs in which every node admits a feature vector. The latter data structure forms the basis of many modern graph learning tasks, such as node classification, graph classification, or graph regression. The predominant class of methods to deal with these graph learning problems are message-passing graph neural networks. We develop an alternative procedure for dealing with featured graph data, built on $\ell$-ECTs and we show that $\ell$-ECTs provide sufficient information to perform message passing, which we explain in the following.

**Definition 1.** *A featured graph is a (non-directed) graph $\mathcal{G}$ such that every node $v \in \mathcal{G}$ admits a feature vector $x(v) \in \mathbb{R}^n$. We denote the set of nodes of $\mathcal{G}$ by $V(\mathcal{G})$, and the set of edges by $E(\mathcal{G})$.*

We notice that a featured graph $\mathcal{G}$ can naturally be interpreted as a graph embedded in $\mathbb{R}^n$, by representing each node feature vector as a point in $\mathbb{R}^n$, and by drawing an edge between two embedded points if and only if there is an edge between the underlying nodes in $\mathcal{G}$. This construction yields a graph isomorphism between $\mathcal{G}$ and the embedded graph if and only if for any pair of nodes $v, w \in \mathcal{G}$ with $v \neq w$ we have $x(v) \neq x(w)$ for their associated feature vectors. In practice, the latter assumption can always be achieved by adding an arbitrarily small portion of Gaussian noise to each feature vector, and we therefore may restrict ourselves to featured graphs that yield an isomorphism on their Euclidean embeddings. We now show that $\ell$-ECTs are in fact expressive representations in the context of graph learning.

**Theorem 2.** *Let $\mathcal{G}$ be a featured graph and let $\{\ell\text{-ECT}_1(x; \mathcal{G})\}_x$ be the collection of local ECTs with respect to the 1-hop neighborhoods in $\mathcal{G}$. Then the collection $\{\ell\text{-ECT}_1(x; \mathcal{G})\}_x$ provides the necessary (non-learnable) information for performing a single message-passing step on $\mathcal{G}$, in the sense that for a given vertex $x \in \mathcal{G}$ one can reconstruct the feature vectors of its 1-hop neighborhood from $\ell\text{-ECT}_1(x; \mathcal{G})$.*

Thm. 2 tells us that for a featured graph $\mathcal{G}$ the collection $\{\ell\text{-ECT}_1(x; \mathcal{G})\}_x$ already contains sufficient information to perform a message passing step. The advantage of using $\ell$-ECTs instead of message passing to represent featured graph data lies in the possibility to additionally use $\{\ell\text{-ECT}_k(x; \mathcal{G})\}_x$ for $k \geq 2$, which contain both structural and feature vector information of larger neighborhoods of nodes in the graph. This type of information is typically *not* explicitly available through message passing since passing messages to non-direct neighbors depends on prior message passing steps, which solely produce an aggregation of neighboring feature vectors.

**Subgraph counting**   Chen et al. (2020) investigate the power of message-passing neural networks with respect to subgraph counting. Specifically, it is proven that message-passing neural networks

*cannot* perform induced-subgraph-count of any connected substructure consisting of 3 or more nodes. By contrast, we will now show that ECTs for featured graphs and their local variants can indeed be used to perform subgraph counting. We start with the definitions of the necessary concepts.

**Definition 2.** *Two featured graphs $\mathcal{G}_1$ and $\mathcal{G}_2$ are isomorphic if there is a bijection $\pi : V(\mathcal{G}_1) \to V(\mathcal{G}_2)$, such that $(v, w) \in E(\mathcal{G}_1)$ if and only if $(\pi(v), \pi(w)) \in E(\mathcal{G}_2)$ and so that for all $v \in \mathcal{G}_1$ one has $x(v) = x(\pi(v))$ for the respective feature vectors.*

A featured graph $\mathcal{G}_S$ is called a subgraph of $\mathcal{G}$ if $V(\mathcal{G}_S) \subset V(\mathcal{G})$ and $E(\mathcal{G}_S) \subset E(\mathcal{G})$, and such that the respective node features remain unaltered under the induced embedding. A featured graph $\mathcal{G}_S$ is called an induced subgraph of $\mathcal{G}$, if $\mathcal{G}_S$ is a subgraph of $\mathcal{G}$, and if $E(\mathcal{G}_S) = E(\mathcal{G}) \cap \mathcal{G}_S$. For two featured graphs $\mathcal{G}$ and $\mathcal{G}_S$, we define $C_{Sub}(\mathcal{G}; \mathcal{G}_S)$ to be the number of subgraphs in $\mathcal{G}$ that are isomorphic to $\mathcal{G}_S$. Similarly, we define $C_{Ind}(\mathcal{G}; \mathcal{G}_S)$ to be the number of induced subgraphs in $\mathcal{G}$ which are isomorphic to $\mathcal{G}_S$.

**Theorem 3.** *Two featured graphs $\mathcal{G}_1$ and $\mathcal{G}_2$ are isomorphic if and only if $\mathrm{ECT}(\mathcal{G}_1) = \mathrm{ECT}(\mathcal{G}_2)$.*

An immediate consequence of the previous Theorem is the following:

**Corollary 1.** ECT*s can perform subgraph counting.*

As previously stated, GNNs can generally not be used to perform (induced) subgraph counting, and we therefore conclude that ECT-based methods for graph representation learning can be more powerful than message-passing-based approaches.

**Rotation-invariant metric based on local ECTs**   The aforementioned invariance properties of ECTs with respect to translations and scalings naturally raise the question if $\ell$-ECTs may be used to compare the local neighborhoods of two distinct points. Unfortunately, the ECT is sensitive to rotations since rotating the underlying simplicial complex leads to a misalignment of the respective directions in $S^{n-1}$. Since a local comparison should not depend on the choice of a coordinate system, this property is a fundamental obstruction of using $\ell$-ECT as a local similarity measure. We therefore construct a rotation-invariant metric as follows. Let $X, Y \subset \mathbb{R}^n$ be two finite simplicial complexes. Since $X, Y$ are finite, $\mathrm{ECT}(X)$ and $\mathrm{ECT}(Y)$ only take finitely many values, and we can therefore define a similarity measure $d_{\mathrm{ECT}}$ between $X$ and $Y$ as

$$d_{\mathrm{ECT}}(X, Y) := \inf_{\rho \in SO(n)} \|(\mathrm{ECT}(X) - \mathrm{ECT}(\rho Y))\|_\infty \tag{6}$$

We first prove that this similarity measure satisfies the definitions of a metric.

**Theorem 4.** $d_{\mathrm{ECT}}$ *is a metric on the collection of rotation classes of finite simplicial complexes embedded in $\mathbb{R}^n$.*

Thm. 4 ensures that we may use $d_{\mathrm{ECT}}$ as a metric that measures the similarity between embedded simplicial complexes up to rotation. In particular, for a simplicial complex $X \subset \mathbb{R}^n$ and $x, y \in X$, we have a rotation-invariant measure to compare local neighborhoods of $x$ and $y$ by setting

$$d_{\mathrm{ECT}}^k(x, y; X) := \inf_{\rho \in SO(n)} \|\ell\text{-}\mathrm{ECT}_k(x; X) - \ell\text{-}\mathrm{ECT}_k(y; \rho X)\|_\infty \tag{7}$$

In practice, we approximate $d_{\mathrm{ECT}}$ by

$$d_{\mathrm{ECT}}(X, Y) \approx \inf_{\rho \in SO(n)} \left\|\overline{\mathrm{ECT}}(X)_{(m,l)} - \overline{\mathrm{ECT}}(\rho Y)_{(m,l)}\right\|_\infty \tag{8}$$

for a choice of samples $v_1, \ldots, v_m \in S^{n-1}$ and $t_1, \ldots, t_l \in \mathbb{R}$ (and in an analogous manner for the local version $d_{\mathrm{ECT}}^k(x, y; X)$). As we pointed out before, the approximations of the ECTs used in Eq. 8 have a natural vector representation, so that the $\|\bullet\|_\infty$ in Eq. 8 is in fact the maximum of the entry-wise absolute differences between the two respective representation vectors. Therefore, the approximation shown in Eq. 8 is efficiently computable for a large class of data spaces, however for our experiments in Sec. 5 we use the Euclidean metric instead for differentiability reasons.

**Limitations**   While $\ell$-ECTs present clear advantages in preserving *local* details, there are some trade-offs to consider. In certain cases, message-passing GNNs, which aggregate information across neighbors, may be preferable for tasks where global context is more important than local details. Furthermore, while our method is computationally feasible on medium-sized datasets (as demonstrated in our experiments), the complexity of "naïvely" calculating $\ell$-ECTs increases for larger $k$ and with the size and density of the graph, suggesting a need for improved methods.

# 5 EXPERIMENTS

In this section, we present experiments to empirically evaluate the performance of the $\ell$-ECT-based approach in graph representation learning, focusing on node classification tasks. We aim to demonstrate how $\ell$-ECT representations can capture structural information more effectively than traditional message-passing mechanisms, especially in scenarios with high heterophily. Our experiments compare the performance of $\ell$-ECT-based models against the standard GNN models graph attention networks (GATs), graph convolutional networks (GCNs), graph isomorphism network (GIN) as well as the heterophily-specific architecture H2GCN (Zhu et al., 2020). Furthermore, we showcase how the rotation-invariant metric from Sec. 4 may be used for spatial alignment of graph data through experiments.

## 5.1 $\ell$-ECTS IN GRAPH REPRESENTATION LEARNING

The link between message passing and $\ell$-ECTs, which we stated in Thm. 2, encourages us to empirically validate the expressivity of $\ell$-ECTs in the context of node classification. For a featured graph $\mathcal{G}$ and fixed $k \geq 0$, we assign $\ell$-ECT$_k(x; \mathcal{G})$ to every node $v \in \mathcal{G}$. The $\ell$-ECT corresponding to a node together with the respective node feature vector is subsequently used as the input of a simple model for the underlying node classification task. In our experiments, this classifier is given by XGBoost, as we found it to outperform other, more complex models like neural networks, but we emphasize that the choice of the model can be controlled by the user. Notice that we do *not* claim to have found a new state-of-the-art for benchmarking graph datasets, but we rather showcase that an approach based on $\ell$-ECT lets us obtain results that are *on a par with and often superior* to common graph-learning techniques based on message passing, like graph attention networks (GATs) and graph convolutional networks (GCNs).

We find that $\ell$-ECTs work particularly well in situations where aggregating neighboring information is inappropriate, as for graphs that exhibit a high degree of heterophily, for example. In these situations our approach may outperform message-passing-based methods by far. The upshot of our method is that local graph information can be incorporated without the architectural necessity to diffuse information along the graph structure, as it is the case for message-passing-based models. While this discrete diffusion process induced by message passing is useful for a plethora of graph learning tasks, it can also be an obstruction in learning the right representation for tasks where node features of neighbors in the graph should not be aggregated. In this sense, $\ell$-ECTs naturally overcome a fundamental limitation, which (by design) is induced by message passing. The second advantage of $\ell$-ECTs is that they are agnostic to the choice of the downstream model. This allows using models that are easy to tune, enabling practitioners to make use of their graph data without necessarily having specialized knowledge in GNN training and tuning. Moreover, it permits using models that are *interpretable*, making our method well-suited for domains where regulatory demands often ask for levels of interpretability that cannot readily be achieved by (graph) neural networks. In fact, by using feature importance values (which are directly available for tree-based algorithms like XGBoost) and since the entries of the $\ell$-ECT vectors that are used as the input for the model can be linked to the directions in the calculation of the $\ell$-ECTs, one obtains a deeper spatial understanding of how the model arrives at predictions. A further discussion on interpretability and an ablation on the directions used for the $\ell$-ECTs is contained in the appendix. In particular, this shows the expressivity of $\ell$-ECTs for node classification tasks as removing directions leads to lower performance, in general. Finally, we notice that our method enables practitioners to make use of models that do not necessitate a validation split in the dataset, making a larger portion of the data accessible for training and testing.

**Implementation details** For our experiments, we assume that we are given a featured graph $\mathcal{G}$ such that there is an assignment $V(\mathcal{G}) \to \mathcal{Y}$, with $V(\mathcal{G})$ being the node set of $\mathcal{G}$ and $\mathcal{Y}$ being the space of classes w.r.t. the underlying node classification task. For a fixed $k \geq 0$, $x \in V(\mathcal{G})$ and $N_k(x; \mathcal{G})$ being the $k$-hop neighborhood of $x$ in $\mathcal{G}$, we then approximate $\ell$-ECT$_k(x; \mathcal{G})$ via $\overline{\mathrm{ECT}}(N_k(x; \mathcal{G}))_{(m,l)}$ for sampled directions and filtration steps, as explained in Sec.4. In our setup we use $m = l = 64$, but the number of samples can be seen as hyperparameters and may be tuned in practice[1]. The resulting $m \cdot l$-dimensional vector(s) $\overline{\mathrm{ECT}}(N_k(x; \mathcal{G}))_{(m,l)}$ together with the feature

---

[1]Thm.1 provides an order of magnitude for a reasonable choice of $m$ and $l$.

vector of $x$ then serve as additional input information for the classifier, corresponding to node $x$. As the classifying model we use a simple XGBoost classifier without tuning the hyperparameters, which should showcase how well our method performs out of the box. The architecture of our baseline models is mostly inspired by Platonov et al. (2023). In particular, we add a two-layer MLP after every graph neighborhood aggregation layer and further augment the models with skip connections (He et al., 2016) and layer normalization (Ba et al., 2016), for the standard architectures GCN and GAT. For each run, the respective model is trained for 1000 epochs, and we report the test accuracy corresponding to the state of the model that admits the maximum validation accuracy during training. The performance of our baseline models are comparable with the results stated in Platonov et al. (2023).

We start our observations with the WebKB datasets, first introduced in Pei et al. (2020). For all three datasets Cornell, Wisconsin and Texas our $\ell$-ECT-based approach outperforms the baseline GNNs, by far. While the combination of both $\ell$-ECT$_1$ and $\ell$-ECT$_2$ performs best for Cornell and Texas, using solely $\ell$-ECT$_1$ leads to best performance for Wisconsin. However, also for the two aforementioned datasets the combination of $\ell$-ECT$_1$ and $\ell$-ECT$_2$ only slightly improves the performance in comparison to $\ell$-ECT$_1$, suggesting that 1-hop neighboring information is already very informative for these tasks. The results are summarized in Tab. 1.

| Model | Cornell | Wisconsin | Texas |
|---|---|---|---|
| GCN | $45.0 \pm 2.2$ % | $44.2 \pm 2.6$ % | $47.3 \pm 1.5$ % |
| GAT | $44.7 \pm 2.9$ % | $48.2 \pm 2.0$ % | $51.7 \pm 3.2$ % |
| GIN | $46.5 \pm 3.1$ % | $49.7 \pm 2.5$ % | $54.2 \pm 2.9$ % |
| H2GCN | $66.2 \pm 3.5$ % | $70.2 \pm 2.3$ % | $72.3 \pm 3.0$ % |
| $\ell$-ECT$_1$ | $66.8 \pm 4.2$ % | $\mathbf{81.2 \pm 2.9}$ % | $74.6 \pm 0.5$ % |
| $\ell$-ECT$_2$ | $67.0 \pm 4.9$ % | $76.1 \pm 2.8$ % | $73.8 \pm 2.6$ % |
| $\ell$-ECT$_1 + \ell$-ECT$_2$ | $\mathbf{67.1 \pm 4.1}$ % | $78.5 \pm 2.6$ % | $\mathbf{74.8 \pm 3.1}$ % |

Table 1: Performance (in accuracy) of different graph learning models across multiple datasets, for 5 training runs each. The results for the $\ell$-ECTs are with respect to a simple XGBoost classifier.

The heterophilous graph datasets introduced in Platonov et al. (2023) contain the two multiclass graph datasets Roman Empire and Amazon Ratings on which we validate our method. Again, $\ell$-ECT$_1 + \ell$-ECT$_2$ performs best on both datasets, and outperforms the baseline models significantly (see Tab. 2). The results are closely aligned with the observation made in Platonov et al. (2023) that specialized architectures like H2GCN often perform less accurate than standard architectures. Moreover, $\ell$-ECT$_1$ outperforms $\ell$-ECT$_2$ on Roman Empire, while $\ell$-ECT$_2$ outperforms $\ell$-ECT$_1$ on Amazon Ratings. Our interpretation of this finding is that the 1-hop neighborhoods are particularly informative for Roman Empire, while the 2-hop neighborhoods are more informative for the Amazon Ratings dataset.

| Model | Roman Empire | Amazon Ratings |
|---|---|---|
| GCN | $73.3 \pm 0.8$ % | $42.3 \pm 0.7$ % |
| GAT | $76.4 \pm 1.2$ % | $44.6 \pm 0.9$ % |
| GIN | $56.8 \pm 1.0$ % | $44.1 \pm 0.8$ % |
| H2GCN | $64.2 \pm 0.9$ % | $40.1 \pm 0.7$ % |
| $\ell$-ECT$_1$ | $80.4 \pm 0.4$ % | $48.4 \pm 0.3$ % |
| $\ell$-ECT$_2$ | $78.0 \pm 0.3$ % | $49.6 \pm 0.3$ % |
| $\ell$-ECT$_1 + \ell$-ECT$_2$ | $\mathbf{81.1 \pm 0.4}$ % | $\mathbf{49.8 \pm 0.3}$ % |

Table 2: Performance (in accuracy) of different graph learning models across multiple datasets, for 5 training runs each. The results for the $\ell$-ECTs are with respect to a simple XGBoost classifier.

The Amazon dataset was first introduced in Shchur et al. (2018) and consists of the two datasets Computers and Photo. While GAT outperforms on Computers, the combination of $\ell$-ECT$_1$ and $\ell$-ECT$_2$ outperforms on Photo, see Tab.3. However, both of the aforementioned models lead to very similar performance on these two datasets.

| Model | Computers | Photo |
|---|---|---|
| GCN | $91.6 \pm 1.6$ % | $93.6 \pm 1.7$ % |
| GAT | $\mathbf{92.4 \pm 1.3}$ % | $94.8 \pm 1.1$ % |
| GIN | $55.9 \pm 1.5$ % | $82.2 \pm 1.3$ % |
| H2GCN | $84.5 \pm 1.4$ % | $92.8 \pm 1.2$ % |
| $\ell$-ECT$_1$ | $89.6 \pm 0.3$ % | $94.1 \pm 0.3$ % |
| $\ell$-ECT$_2$ | $90.1 \pm 0.5$ % | $94.4 \pm 0.7$ % |
| $\ell$-ECT$_1 + \ell$-ECT$_2$ | $92.2 \pm 0.6$ % | $\mathbf{94.9 \pm 0.6}$ % |

Table 3: Performance (in accuracy) of different graph learning models across multiple datasets, for 5 training runs each. The results for the $\ell$-ECTs are with respect to a simple XGBoost classifier.

For the Actor dataset (Pei et al., 2020), the $\ell$-ECT$_1$ model achieves the highest accuracy, outperforming all other baselines. The combination $\ell$-ECT$_1 + \ell$-ECT$_2$ performs slightly worse. $\ell$-ECT$_2$ performs the lowest on this dataset, indicating that $\ell$-ECT$_1$ captures more relevant features for the Actor dataset. The WikipediaNetwork dataset from Rozemberczki et al. (2021) consists of the two node classification tasks Squirrel and Chameleon. On both datasets, the heterophily-specific architecture H2GCN performs best. However, $\ell$-ECT$_1$ and the combination of $\ell$-ECT$_1$ and $\ell$-ECT$_2$ lead to similar or even better accuracies as all other standard baselines, showcasing that our proposed method performs surprisingly well out of the box.

| Model | Actor | Squirrel | Chameleon |
|---|---|---|---|
| GCN | $30.7 \pm 2.1$ % | $28.9 \pm 1.4$ % | $42.8 \pm 1.8$ % |
| GAT | $31.1 \pm 1.8$ % | $31.8 \pm 1.3$ % | $47.3 \pm 1.3$ % |
| GIN | $26.5 \pm 2.0$ % | $35.4 \pm 1.5$ % | $43.1 \pm 1.7$ % |
| H2GCN | $30.7 \pm 1.9$ % | $\mathbf{40.8 \pm 1.4}$ % | $\mathbf{62.7 \pm 1.6}$ % |
| $\ell$-ECT$_1$ | $\mathbf{31.4 \pm 1.9}$ % | $35.6 \pm 0.7$ % | $43.46 \pm 1.7$ % |
| $\ell$-ECT$_2$ | $30.1 \pm 1.3$ % | $35.6 \pm 0.8$ % | $40.44 \pm 1.5$ % |
| $\ell$-ECT$_1 + \ell$-ECT$_2$ | $30.9 \pm 0.7$ % | $35.3 \pm 1.5$ % | $43.90 \pm 0.7$ % |

Table 4: Performance (in accuracy) of different graph learning models across multiple datasets, for 5 training runs each. The results for the $\ell$-ECTs are with respect to a simple XGBoost classifier. The Actor dataset is separated visually to indicate it belongs to a different dataset. The maximum accuracy for each dataset is highlighted in bold.

An evaluation of the experimental results shown in this subsection is presented in App.A.2.3, indicating that $\ell$-ECT-based approaches provide a versatile general-purpose solution for node classification, and demonstrating the capability to handle heterophilic datasets effectively without any additional customization.

## 5.2 LEARNING SPATIAL ALIGNMENT

In the following, we use the approach described in Sec. 4 in order to learn the spatial alignment of two data spaces by re-rotating one into the other. We start by showing that synthetic data which only differs up to a rotation can be re-aligned using the given method. Moreover, we show that this alignment is stable with respect to noise, making it a robust measure for the comparison of local neighborhoods in data. In comparison to other spatial alignment methods like the iterative closest point algorithm, ours does *not* necessitate the computation of all pairwise distances between points in the respective spaces. The latter is often a computational bottleneck, especially for large datasets, thus positioning our method for spatial alignment as a computationally more efficient method in practice. We observe that this approach is also capable of aligning embedded graph data, making it particularly useful for dealing with *geometric graphs*, constituting a highly-efficient alternative to more involved machine-learning models such as geometric GNNs (Joshi et al., 2023). For a discussion of spatial alignment in the context of high-dimensional point cloud data, please refer to App. A.2.2.

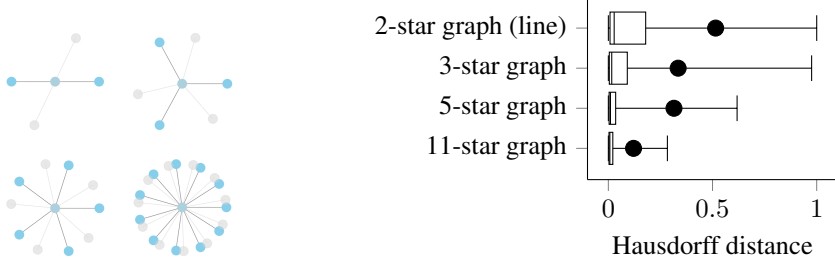

Figure 1: A comparison of the Hausdorff distances of aligned graphs. The black dots represents the Hausdorff distance between the original graph and a randomly rotated version of itself.

For the subsequent learning problem, let us assume that we are given two point clouds $X, Y \subset \mathbb{R}^n$. In light of Sec. 4, the metric properties of $d_{\text{ECT}}$ ensure that $d_{\text{ECT}}(X, Y) = 0$ if $X$ and $Y$ only differ up to a rotation. We therefore approximate $d_{\text{ECT}}(X, Y)$ via the learning problem

$$\min_{\rho \in SO(n)} \left\| \overline{\text{ECT}}(X)_{(m,l)} - \overline{\text{ECT}}(\rho Y)_{(m,l)} \right\|_2^2 \tag{9}$$

for a choice of directions $v_1, \ldots, v_m \in S^{n-1}$ and filtration steps $t_1, \ldots, t_l \in \mathbb{R}$. As explained in Sec. 4, the ECT approximations are given by vectors, making it feasible to approach the above learning problem by a general-purpose gradient-based learning algorithm. The advantage of this formulation is that it yields both the rotation-invariant loss and the rotation that leads to this minimum loss, where the latter is given by the parameters of the learner.

**Low-dimensional data**    We now approximate the optimization problem in Eq. 9 to show that we can learn a spatial alignment of two given data spaces, while the distance between ECTs of non-aligned spaces that only differ up to a rotation will in fact generally be high. Our first example is given by a wedged sphere, meaning two 2-dimensional spheres which are concatenated at a gluing point (see Fig. 3). We begin by sampling 2000 points from such a wedged sphere, and compare the squared L2 loss between the ECTs of this sample and a rotation of the same data space. We repeat this procedure 500 times, where at each step both the sample of the wedged sphere and the rotation matrix which yields the rotated version of the same space are randomly sampled. We notice that the L2 losses between the non-aligned spaces are high (with a median of around 19), whereas the L2 losses of the non-aligned spaces are significantly lower, with a median loss close to zero (see Fig. 8 in the appendix for more details). Moreover, we see that the ECT of the same space significantly changes when the coordinate system is changed, which undermines the necessity of a rotation-invariant metric for a comparison of ECTs that we introduced in Sec. 4. We conclude that an alignment of the ECTs of the two underlying data spaces in fact leads to an alignment of the data spaces itself, as promised by the theoretical results in Sec. 4.

**Robustness with respect to outliers and noise**    Fig. 4 and Fig. 5 in the appendix show that the spatial alignment of wedged spheres still works in the presence of outliers and noise. This property is an important feature when dealing with real-world data, which is often noisy, and enables us to align spaces that only approximately differ up to a rotation. By contrast, the Hausdorff distance which is a widely used metric between point clouds is (by definition) highly sensitive to outliers. We therefore conclude that the proposed metric based on ECTs is a robust metric to compare point clouds of potentially different cardinalities.

**Geometric graphs**    As a final example of the utility of our method, we face the problem of graph re-alignment. Recall that a $k$-star graph is given by a tree with one internal node and $k$ leaves (see Fig. 1 for examples of different $k$-star graphs). In order to obtain an embedded graph, we assign a 2D vector to every node in the graph in such a way that the assigned node vectors are equidistant to each other. Subsequently, the embedded graph is perturbed by a random 2D rotation, and finally the rotation matrix is learned by using the ECT-based metric defined in Sec. 4. We measure the similarity via the Hausdorff distance between the original graph and its re-rotated version. We repeated the learning procedure for 200 times with the same initialization of both the graph and the respective rotation matrix. The results are shown in Fig. 1. We see that the realignment leads to

small Hausdorff distances with medians close to zero, whereas the Hausdorff distance between the original graph and its random perturbation is significantly higher.

# 6 DISCUSSION

In this work, we introduced the Local Euler Characteristic Transform ($\ell$-ECT), providing a novel approach to graph representation learning that preserves local structural information without relying on aggregation. Our method addresses fundamental limitations in message-passing neural networks, particularly in tasks where aggregating neighboring information is suboptimal, such as in graphs with heterophily. By retaining critical local details, $\ell$-ECTs enable more nuanced and expressive representations, offering significant advantages in node classification tasks and beyond.

One key strength of our approach is its model-agnostic nature, allowing it to be paired with interpretable machine learning models. This is particularly useful in domains such as healthcare, finance, and legal applications, where regulatory frameworks demand high levels of transparency and interpretability that are often difficult to achieve with black-box neural networks. By leveraging $\ell$-ECTs, we can fulfill these regulatory requirements while maintaining the expressiveness needed for accurate representation learning.

Beyond graph representation learning, the framework of $\ell$-ECTs opens the door for applications in other areas where local structure is critical, such as in point clouds, 3D shape analysis, and biological networks. Additionally, the ability to generalize $\ell$-ECTs to higher-dimensional simplicial complexes suggests future extensions of this work into the realm of simplicial learning, providing a powerful tool for analyzing higher-order data in a computationally feasible way. Future work could explore more efficient algorithms for computing $\ell$-ECTs at scale, as well as hybrid approaches that balance local and global information more effectively. In particular, the investigation of sampling methods appears to be a promising direction for this purpose. Moreover, heterophily-specific mechanisms such as separation of neighborhood aggregation (as used for specialized GNN architectures) may be incorporated into our $\ell$-ECT-based framework to further strengthen its expressivity in the presence of high-heterophily graphs.

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

## A  APPENDIX

### A.1  PROOFS

**Proof of Theorem 1:**  In the setting of Sec.4, the worst case convergence of $\overline{\mathrm{ECT}}(X)_{(m,l)}$ to the true quantity $\mathrm{ECT}(X)$ is $\mathcal{O}((\frac{\log m}{m})^{1/(n-1)}\frac{1}{l})$.

*Proof.* The total surface area of $S^{n-1}$ is given by $A(S^{n-1}) = \frac{2\pi^{n/2}}{\Gamma(n/2)}$. A spherical cap on $S^{n-1}$ is the set of points on the sphere that lie within an angular distance $\delta$ from a given point, and the area of such a spherical cap can for large $m$ be approximated as $A_{\mathrm{cap}} \approx c_n \delta^{n-1}$ with $c_n$ being a constant (and where "$\approx$" denotes asymptotic equivalence). What we would like to achieve is $A_{\mathrm{cap}} \approx \frac{A(S^{n-1})}{m}$, leading to $\delta \approx c'_n(\frac{1}{m})^{1/(n-1)}$ for a constant $c'_n$. However, the discrepancy in the uniform sampling on the sphere introduces another logarithmic term, leading to $\delta \approx c''_n(\frac{logm}{m})^{1/(n-1)}$, see Beck (1987). The convergence speed with respect to the Euclidean direction is $\mathcal{O}(\frac{1}{l})$ since no sampling is involved here (as we use an equidistant partitioning of the respective interval). Together, this proves the statement. $\qquad\square$

**Proof of Theorem 2:**  Let $\mathcal{G}$ be a featured graph and let $\{\ell\text{-}\mathrm{ECT}_1(x;\mathcal{G})\}_x$ be the collection of local ECTs with respect to the 1-hop neighborhoods in $\mathcal{G}$. Then the collection $\{\ell\text{-}\mathrm{ECT}_1(x;\mathcal{G})\}_x$ provides the necessary information for performing a single message-passing step on $\mathcal{G}$, in the sense that for a given vertex $x \in \mathcal{G}$ one can reconstruct the feature vectors of its 1-hop neighborhood from $\ell\text{-}\mathrm{ECT}_1(x;\mathcal{G})$.

*Proof.* By the remark before, we may assume that the natural embedding of $\mathcal{G}$ into $\mathbb{R}^n$ is a graph isomorphism. Then by the invertibility theorem, the 1-hop neighborhood of a point $x$ in the embedding of $\mathcal{G}$ can be reconstructed from $\ell\text{-}\mathrm{ECT}_1(x;\mathcal{G})$. Therefore, the feature vectors of $x$ and its 1-hop neighbors can be deduced from $\ell\text{-}\mathrm{ECT}_1(x;\mathcal{G})$, which is the only non-learnable information one needs to perform a message passing step. $\qquad\square$

**Proof of Theorem 3:**  Two featured graphs $\mathcal{G}_1$ and $\mathcal{G}_2$ are isomorphic if and only if $\mathrm{ECT}(\mathcal{G}_1) = \mathrm{ECT}(\mathcal{G}_2)$.

*Proof.* When two featured graphs are isomorphic in the sense of Def.2, their respective Euclidean embeddings produce equal ECTs, by construction. On the other hand, let us assume that $\mathrm{ECT}(\mathcal{G}_1) = \mathrm{ECT}(\mathcal{G}_2)$. Then by the invertibility theorem, the Euclidean embeddings of $\mathcal{G}_1$ and $\mathcal{G}_2$ are equal. Therefore, the only information that may tell apart the two graphs are their node labels, but this means that $\mathcal{G}_1$ and $\mathcal{G}_2$ are isomorphic. $\qquad\square$

**Proof of Theorem 4:**  $d_{\mathrm{ECT}}$ is a metric on the collection of rotation classes of finite simplicial complexes embedded in $\mathbb{R}^n$.

*Proof.* $d_{\mathrm{ECT}}(X,X) = 0$ holds for $\rho$ being the identity. Now assume that $d_{\mathrm{ECT}}(X,Y) = 0$. Then there exists $\rho \in SO(n)$ with $\|(\mathrm{ECT}(X) - \mathrm{ECT}(\rho Y))\|_\infty = 0$. As $\|\bullet\|_\infty$ is a norm, it follows that $\mathrm{ECT}(X) = \mathrm{ECT}(\rho Y)$, and by the invertibility theorem we obtain $X = \rho Y$. This shows the first property of a metric (note that positivity follows from $\|\bullet\|_\infty$). For symmetry, note

that $\left\|(\mathrm{ECT}(X) - \mathrm{ECT}(\rho Y))\right\|_\infty = \left\|(\mathrm{ECT}(\rho^{-1}X) - \mathrm{ECT}(Y))\right\|_\infty$, since rotations are invertible. For the triangle inequality, let $Z$ be another finite simplicial complex. We then have

$$
\begin{aligned}
d_{\mathrm{ECT}}(X,Z) &= \inf_{\rho \in SO(n)} \left\|(\mathrm{ECT}(X) - \mathrm{ECT}(\rho Z))\right\|_\infty \\
&\leq \inf_{\rho,\rho' \in SO(n)} \left\|(\mathrm{ECT}(X) - \mathrm{ECT}(\rho'Y))\right\|_\infty + \left\|(\mathrm{ECT}(\rho'Y) - \mathrm{ECT}(\rho Z))\right\|_\infty \\
&= \inf_{\rho,\rho' \in SO(n)} \left\|(\mathrm{ECT}(X) - \mathrm{ECT}(\rho'Y))\right\|_\infty + \left\|(\mathrm{ECT}(Y) - \mathrm{ECT}((\rho')^{-1}\rho Z))\right\|_\infty \\
&= \inf_{\rho \in SO(n)} \left\|(\mathrm{ECT}(X) - \mathrm{ECT}(\rho Y))\right\|_\infty + \inf_{\rho \in SO(n)} \left\|(\mathrm{ECT}(Y) - \mathrm{ECT}(\rho Z))\right\|_\infty \\
&= d_{\mathrm{ECT}}(X,Y) + d_{\mathrm{ECT}}(Y,Z)
\end{aligned}
$$

$\square$

## A.2 Additional results

As another node classification task, we provide results for the well-known Planetoid datasets from Yang et al. (2016) which consists of Cora, CiteSeer and PubMed. We trained all models using a random 75/25 split of the data. The results are shown in Tab.5. Although GCN and GAT performs slightly better for Cora and CiteSeer, the gap is surprisingly small. In the case of PubMed the $\ell$-ECT-based models even outperform both GCN and GAT. The findings suggest that the expressivity of $\ell$-ECTs which we formally established in Sec. 4 is also of practical use, giving rise to an alternative way of dealing with graph data that is not restricted by the underlying model architecture and therefore allows for interpretability.

| Model | Cora | CiteSeer | PubMed |
|---|---|---|---|
| GCN | $88.1 \pm 1.2\ \%$ | $74.6 \pm 1.5\ \%$ | $85.3 \pm 4.7\ \%$ |
| GAT | $\mathbf{88.3 \pm 1.1\ \%}$ | $\mathbf{75.3 \pm 1.5\ \%}$ | $85.7 \pm 4.2\ \%$ |
| $\ell$-ECT$_1$ | $87.6 \pm 0.6\ \%$ | $72.1 \pm 0.6\ \%$ | $90.2 \pm 0.5\ \%$ |
| $\ell$-ECT$_2$ | $87.2 \pm 0.7\ \%$ | $72.3 \pm 0.8\ \%$ | $\mathbf{90.3 \pm 0.5\ \%}$ |
| $\ell$-ECT$_1$ + $\ell$-ECT$_2$ | $87.8 \pm 0.6\ \%$ | $72.5 \pm 0.7\ \%$ | $\mathbf{90.3 \pm 0.5\ \%}$ |

Table 5: Performance (in accuracy) of different graph learning models across multiple datasets, for 5 training runs each. The results for the $\ell$-ECTs are with respect to a simple XGBoost classifier.

### A.2.1 Ablation on directions and interpretability

Coming back to our approximation of $\mathrm{ECT}(X)$ via $\overline{\mathrm{ECT}}(X)_{(m,l)} := \mathrm{ECT}(X)_{|\{v_1,\ldots,v_m\} \times \{t_1,\ldots,t_l\}}$ for uniformly distributed directions $v_1,\ldots,v_m \in S^{n-1}$ and filtration steps $t_1,\ldots,t_l \in \mathbb{R}$, we notice that the $(l \cdot (j-1)+1)$-th till $(l \cdot j)$-th entries of $\overline{\mathrm{ECT}}(X)_{(m,l)}$ correspond to the direction $v_j$. The latter gives us the opportunity to get a deeper spatial understanding of how the model predicts its outcome, by analysing its feature importances (which are available for tree-based algorithms like e.g. XGBoost). Therefore, our approach enables us to analyse which features (i.e. directions) of the underlying ECT vector are most important. In practice, we often observe that a small number of features admits high feature imoprtance with respect to the corresponding model, see Fig.2. This raises the question if we may use only a smaller random collection of features and still get reasonable results. We therefore ran experiments for a collection of datasets for a varying number of randomly sampled entries of the $\ell$-ECT$_1$ vector see Tab.6 for the results. Here, 4096 corresponds to the whole vector. We observe that for certain tasks like e.g. Coauthor CS, Coauthor Physics and Amazon Ratings the performance of the model only slightly changes when using only a small portion of the $\ell$-ECT$_1$ vector. In light of the results in Curry et al. (2022), this observation is not entirely surprising: one main claim therein is that the ECT can be determined using only a small number of directions.

### A.2.2 Spatial Alignment of High-Dimensional Data

Following our previous observations that $d_{\mathrm{ECT}}$ enables us to align two spaces, we now use it to investigate its effect on high-dimensional data. We start this discussion with the well-known MNIST

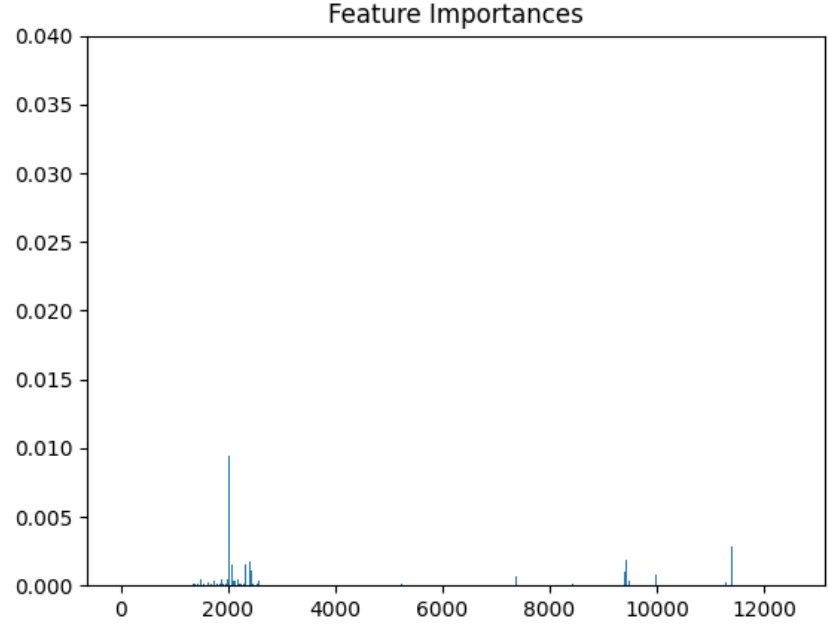

Figure 2: Feature importances of an XGBoost model for the Coauthor Phyiscs dataset (using $\ell$-ECT$_1$). Only a small number of features admit high importance scores.

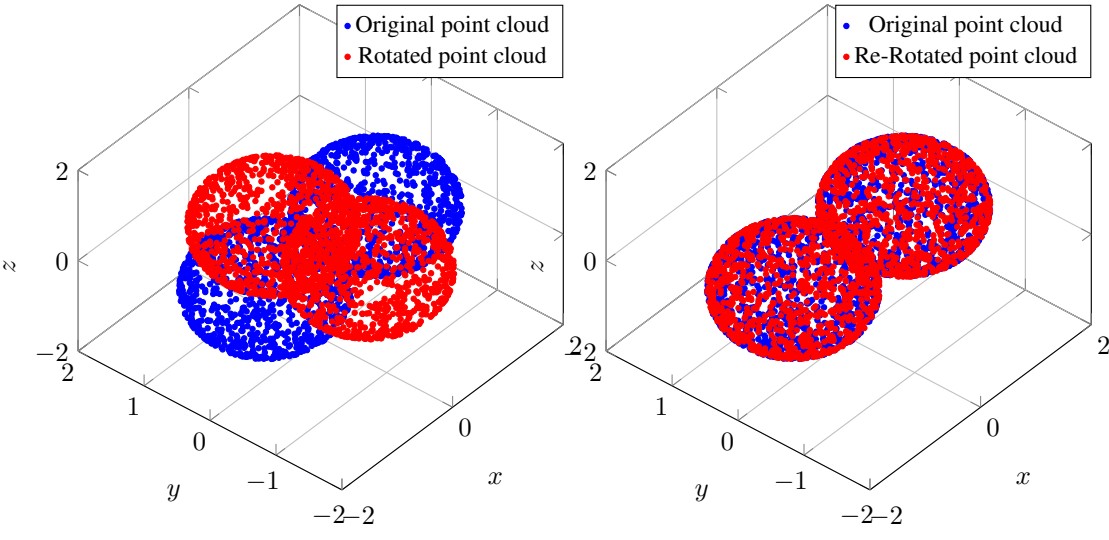

Figure 3: A comparison of two wedged spheres with one being rotated around the wedge point and the points being perturbed by Gaussian noise (left) and the learned re-rotated sphere that is aligned with the original data (right).

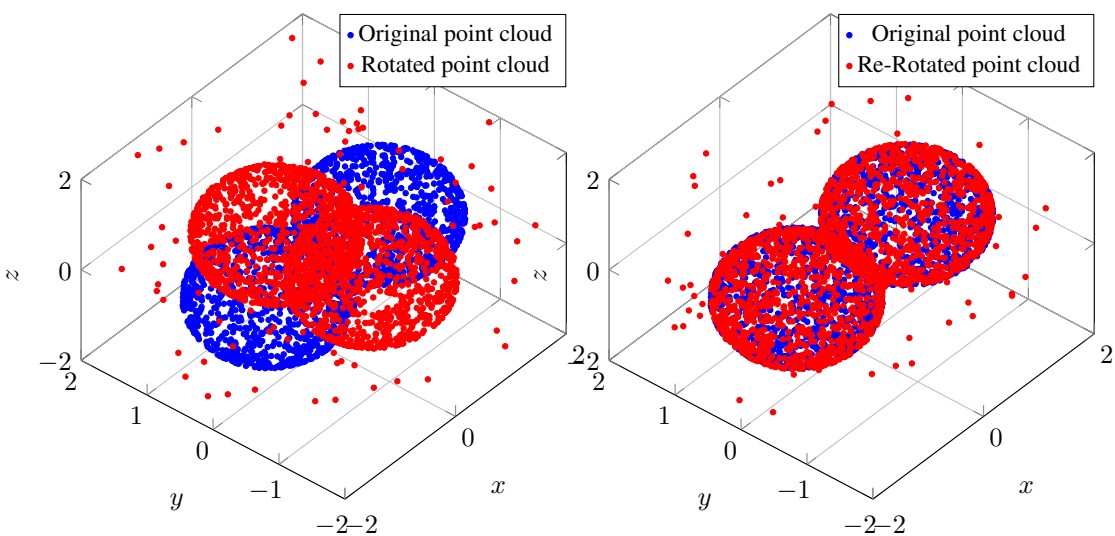

Figure 4: A comparison of two wedged spheres with one being rotated around the wedge point and added 200 outliers (left) and the learned re-rotated sphere that is aligned with the original data (right).

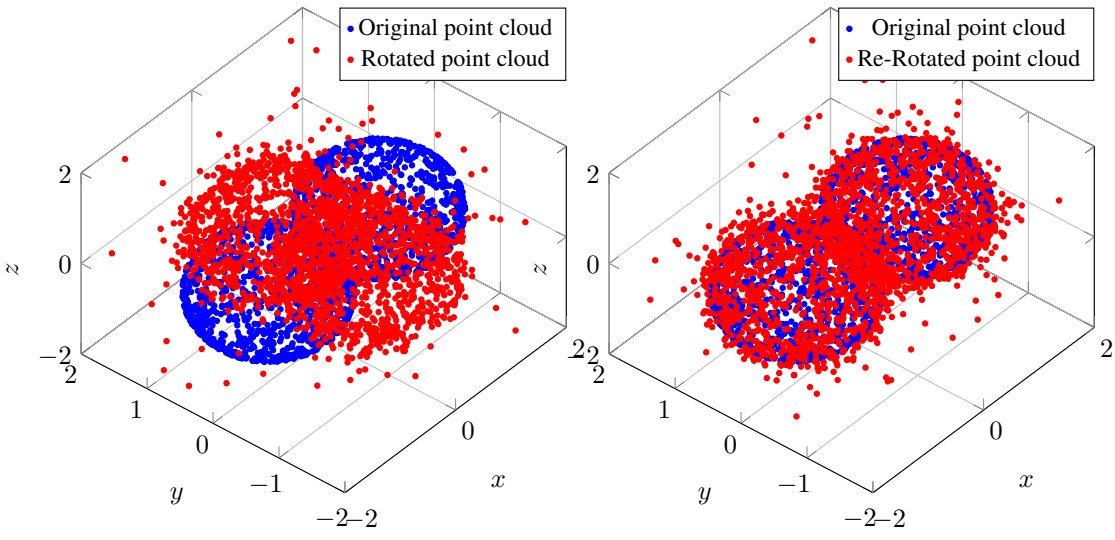

Figure 5: A comparison of two wedged spheres with one being rotated around the wedge point and the points being perturbed by Gaussian noise (left) and the learned re-rotated sphere that is aligned with the original data (right).

Table 6: Mean accuracy results (5 runs each) for different node classification tasks, and a varying number of randomly sampled entries of the respective $\ell$-ECT$_1$ vectors.

| Dataset | 50 | 100 | 500 | 1000 | 4096 |
|---|---|---|---|---|---|
| WikiCS | 69.2% | 70.5% | 71.3% | 72.7% | 74.6% |
| Coauthor CS | 92.3% | 92.4% | 92.5% | 92.6% | 92.6% |
| Coauthor Physics | 95.6% | 95.6% | 95.8% | 95.9% | 96.1% |
| Roman Empire | 73.7% | 75.8% | 78.3% | 79.7% | 80.4% |
| Amazon Ratings | 47.9% | 48.2% | 48.4% | 48.2% | 48.4% |

benchmark dataset. We first represent each (gray-scale) image in the dataset as a 784-dimensional vector, by flattening the image. In this way, we obtain a high-dimensional point cloud corresponding to the dataset. Subsequently, we sample 300 points of digits of '1' and calculate the pairwise distances of their respective $\ell$-ECT (with respect to the whole point cloud), for $k = 10$. Finally, we calculate the pairwise distances of the respective aligned $\ell$-ECTs (by using the approach of Eq. 9 and for $k = 10$). The results are shown in Fig. 6. We see that the aligned $\ell$-ECTs have a significantly lower squared L2 distance (with a median of $\approx 112$) than the non-aligned ones (with a median of $\approx 224$), showcasing that rotations cause dissimilarity between small neighborhoods of points, in many cases.

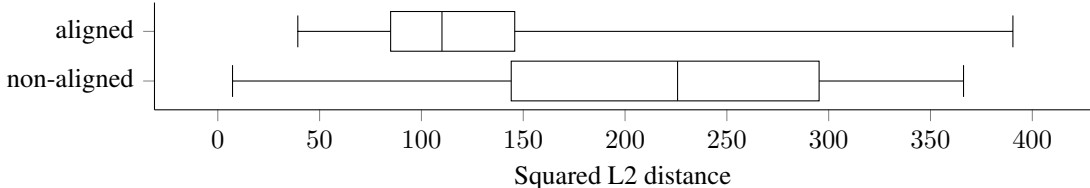

Figure 6: A comparison of the squared L2 distances of $\ell$-ECTs of aligned and non-aligned MNIST digits of '1', respectively.

### A.2.3 POST-HOC EVALUATION OF NODE CLASSIFICATION EXPERIMENTS

A critical difference diagram arranges the average ranks of multiple models across a set of datasets in order to facilitate overall performance comparisons between the model performances. Fig.7 shows the results for all node classification results given in Sec.4 which include both homophilic and heterophilic graph datasets. We see that the $\ell$-ECT-based approaches outperform standard methods and the heterophily-specific architecture H2GCN by far, when averaged over all datasets [2]. The best performing method $\ell$-ECT$_1$ + $\ell$-ECT$_2$ exhibits an average rank of 2, while the worst performing method is GIN with an average rank of 5.7. Even the worst performing $\ell$-ECT-based method ($\ell$-ECT$_2$) performs better than the best non-$\ell$-ECT-based method GAT.

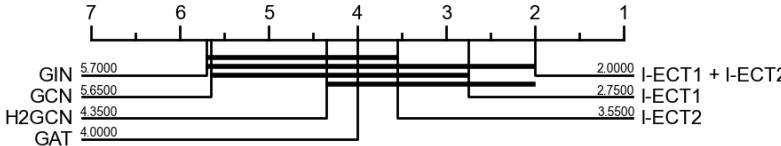

Figure 7: Critical difference diagram for ranks of different models across all node classification tasks in Sec.5. Even the worst performing $\ell$-ECT-bsaed approach ($\ell$-ECT$_2$) works better than all other methods, when averaged over all tasks.

To evaluate the performance of our methods in comparison to those reported in the literature, we included a comparison with the results presented in Platonov et al. (2023), utilizing the ranks of

---

[2]We used `https://github.com/hfawaz/cd-diagram` for the creation of the critical difference diagram.

the respective models as the basis for evaluation. The results are shown in Tab.A.2.3, where the table presents the ranking of various models on heterophilic node classification tasks. Among the listed methods, several, such as H2GCN, CPGNN, and GPR-GNN, are explicitly designed for heterophilic graph settings, leveraging specialized architectures to handle the challenges posed by such data. In contrast, our $\ell$-ECT$_1$ + $\ell$-ECT$_2$ method, despite being a general-purpose approach not tailored specifically for heterophilic settings, achieves a competitive rank of 11. This performance is on par with other top-performing heterophily-specific models, such as GloGNN, and outperforms well-established architectures like GT and GAT by a significant margin. These results highlight the robustness and adaptability of our method, demonstrating its ability to handle diverse graph structures effectively without requiring customization for heterophilic scenarios. In consideration of the results given in Fig.7, this makes $\ell$-ECT-based approaches a versatile general-purpose solution for various node classification tasks.

| Model | Rank |
|---|---|
| H2GCN | 18.250 |
| CPGNN | 16.750 |
| GPR-GNN | 15.250 |
| ResNet | 13.750 |
| l-ECT1 | 12.375 |
| l-ECT2 | 12.375 |
| GAT | 12.250 |
| GT | 11.000 |
| **l-ECT1 + l-ECT** | **11.000** |
| GloGNN | 11.000 |
| ResNet+SGC | 10.750 |
| FAGCN | 10.000 |
| JacobiConv | 9.750 |
| GCN | 9.625 |
| GBK-GNN | 9.000 |
| ResNet+adj | 7.250 |
| SAGE | 5.875 |
| GAT-sep | 5.500 |
| GT-sep | 5.250 |
| FSGNN | 3.000 |

Table 7: Ranks of the various models in Platonov et al. (2023) across the heterophilic datasets therein, in comparison to our methods.

### A.2.4 SPATIAL ALIGNMENT OF WEDGED SPHERES

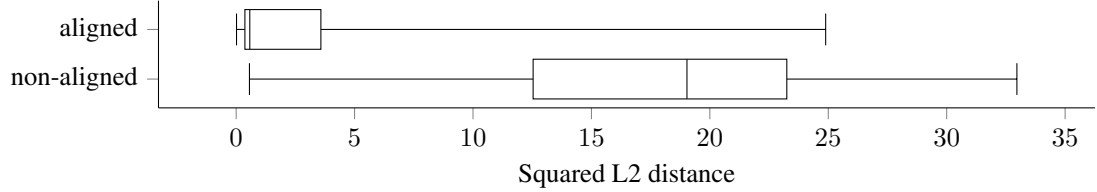

Figure 8: A comparison of the squared L2 distances of the respective ECTs of aligned and non-aligned wedged spheres, respectively.

Fig.8 shows that the L2 losses between the non-aligned spaces are high (with a median of around 19), whereas the L2 losses of the non-aligned spaces are significantly lower, with a median loss close to zero. We therefore see that the ECT of the same space significantly changes when the coordinate system is changed, which undermines the necessity of a rotation-invariant metric for a comparison of ECTs that we introduced in Sec. 4.

**Disclaimer** Certain aspects of this work have been improved with the help of OpenAI (2023).

