# OpenReview forum: "Diss-l-ECT: Dissecting Graph Data with local Euler Characteristic Transforms"
_ICLR.cc/2025/Conference — Submitted to ICLR 2025_

### Official Review · Reviewer_WDbj · 2024-10-17

**Soundness:** 1
**Presentation:** 2
**Contribution:** 2
**Rating:** 3
**Confidence:** 4

**Summary:**

The paper proposes a local Euler characteristic transform for enhancing feature representation for graph learning. This approach addresses key limitations in GNNs by preserving nuanced local structures while maintaining global interpretability.

**Strengths:**

Using local Euler characteristic transform for graph representation is novel to me.

**Weaknesses:**

1. To compute ECT or l-ECT, one needs to embed a simplicial complex in a Euclidean space. The authors propose to embed using node features. However, I don't think this is a genuine embedding. For example, if the feature space is \mathbb{R}^2, then even if the nodes are embedded to the place in a 1-1 fashion, the edges may cross each other. Therefore, only talking about vertex embedding is insufficient as a graph or a simplicial complex has additional structures.
2. Related to 1. The author should be more specific on ``embedding'', whether it is metrical embedding, differential embedding, or topological embedding (or something else?).
3. The proofs are poorly written. The statements are vague and imprecise. Many details are missing. It is hard to assess the correctness of the results.
4. It seems to me that the proposed l-ECTs capture local structural information. They are used as node features, but not used to guide feature aggregation. I fail to get the intuition of why they can be useful for the node classification task. However, on the other hand, they might be useful for the graph classification task.
5. The compared benchmarks are very limited (only GCN and GAT). From my own experience, the results are not very impressive, e.g., for the Actor, Squirrel, and Chameleon datasets, there are more recent benchmarks (e.g., ACM-GCN) whose performance is at least 5%-10% higher than those reported by the authors.
6. Ablation study is missing. It is hard to assess whether l-ECTs play an important role in the reults shown.

**Questions:**

See weaknesses.

---

> ### Author Response · Authors · 2024-11-20
> **Response to your review**
>
> Dear Reviewer,
>
> Thank you for taking your time in reviewing our paper and for acknowledging that our approach addresses key limitations in GNNs by preserving nuanced local structures while maintaining global interpretability.
>
> Regarding your concerns:
>
> > To compute ECT or l-ECT, one needs to embed a simplicial complex in a Euclidean space. The authors propose to embed using node features. However, I don't think this is a genuine embedding. For example, if the feature space is \(\mathbb{R}^2\), then even if the nodes are embedded to the place in a 1-1 fashion, the edges may cross each other. Therefore, only talking about vertex embedding is insufficient as a graph or a simplicial complex has additional structures.
>
> Indeed, embedding only the vertices of the graph/simplicial complex is insufficient as you point out. However, we embed the whole object (as is described in l.162-164) in a way that the additional structure of the object is respected. In the case of graph (on which we focus in the experiments), this means that we embed the graph in a way that we obtain a graph isomorphism on the image of this embedding. Although edge crossings may happen, the invertibility theorem for Euler Characteristic Transforms still ensures that both the feature vector information and the graph structure can be deduced from the respective ECT (note that the crossing point is not a node!).
>
> > Related to 1. The author should be more specific on ``embedding'', whether it is metrical embedding, differential embedding, or topological embedding (or something else?).
>
> By embedding, we mean that we embed the vertices of the simplicial complex using their spatial data (i.e. the feature vectors) and draw edges between embedded nodes in a way that we obtain an isomorphism of simplicial complexes between the original and the embedded complex. In the special case of graphs, this notion of isomorphism is given by graph isomorphism.
> We apologize for the confusion and will clarify this is our revision.
>
> > The proofs are poorly written. The statements are vague and imprecise. Many details are missing. It is hard to assess the correctness of the results.
>
> Thanks for your feedback, we are happy to provide more details on the proofs and revise them accordingly! Could you please pinpoint us to the specific parts which are unclear to you?
>
> > It seems to me that the proposed l-ECTs capture local structural information. They are used as node features, but not used to guide feature aggregation. I fail to get the intuition of why they can be useful for the node classification task. However, on the other hand, they might be useful for the graph classification task.
>
> The l-ECT of a given node can be interpreted as a fingerprint of the ego graph of the respective node. Therefore, l-ECTs capture both feature vector information and the structural information of the local graph neighborhood which is sufficient to restore this neighborhood in a lossless way. In this sense, l-ECTs in fact allow for guiding feature aggregation since feature vectors of neighboring nodes are implicitly contained in the respective l-ECT.
> The intuition is that l-ECTs provide a way to represent local (featured) graph neighborhoods in a lossless way, combining both the structural and spatial information.

---

> > ### Author Response · Authors · 2024-11-20
> > **Response to your review (2/2)**
> >
> > > The compared benchmarks are very limited (only GCN and GAT). From my own experience, the results are not very impressive, e.g., for the Actor, Squirrel, and Chameleon datasets, there are more recent benchmarks (e.g., ACM-GCN) whose performance is at least 5%-10% higher than those reported by the authors.
> >
> > Thank you for pointing this out! Our method is designed as a general-purpose approach, and we deliberately chose widely used baselines like GCN and GAT for comparison since they represent general-purpose graph neural networks that are not explicitly tailored to specific graph settings, such as heterophilic graphs. This aligns with the objective of our method to demonstrate versatility rather than specialize in one domain.
> > We acknowledge the value of including more recent benchmarks, particularly models designed for heterophilic settings, to provide a fuller context. To address your concern, we will include results comparing our method to specialized architectures like ACM-GCN or H2GCN, which are known to perform well on heterophilic datasets. These comparisons will complement the existing evaluations and demonstrate the relative strength of our approach in such settings.
> > Finally, we will provide an extended discussion in the revised paper to contextualize these new results, particularly focusing on the unique contributions of our method as a general-purpose architecture.
> >
> > > Ablation study is missing. It is hard to assess whether l-ECTs play an important role in the reults shown.
> >
> > An ablation on the number of directions used for the l-ECT is contained in the appendix. Since we observe significant improvement with an increasing number of directions, this particularly shows the efficacy of our approach. We will highlight this contribution better in our revision.
> >
> > In the meantime, please feel free to reach out if you have any additional questions. If our responses have sufficiently addressed your concerns, we would be grateful if you could consider re-evaluating your overall rating.
> >
> > Best regards,
> >
> > the Authors

---

> ### Comment · Reviewer_WDbj · 2024-11-23
>
> Thank you for the response to my comments. However, I am not convinced to change my score for the following reasons (hope the remarks can help the authors improve the quality of the paper).
> 1. The issue regarding embedding remains. The authors replied with the following: "we embed the vertices of the simplicial complex using their spatial data (i.e. the feature vectors) and draw edges between embedded nodes in a way that we obtain an isomorphism of simplicial complexes between the original and the embedded complex." However, in general, I believe such "an embedding" will not be a graph isomorphism or a homeomorphism, and likely not even a homotopy equivalence. Hence, many topological properties (e.g., Euler characteristic) may be changed during the process.
> 2. Regarding the proofs, I think it is the responsibility of the authors to make them readable, precise, and rigorous. For example, in the proof of Theorem 1, there are many vague phrasing such as "$\approx$" (how to quantify this?), and the meaning of "since no sampling is involved" is unclear. In the statement of Theorem 2, the statement "provides the necessary information for performing a single message-passing step" is imprecise and not acceptable as a theoretical result. Such issues are all over the place in Appendix A.1.
> 3. Regarding using I-ECT to generate features, I am not convinced that the local structures of nodes are important in distinguishing node classes. There can be nodes with the same label but completely different neighborhood structures. However, the local topological structures might be useful for the graph classification task.
> 4. Numerical studies are insufficient as I have pointed out in the review, and they are not fully addressed in the rebuttal.

---

> > ### Author Response · Authors · 2024-11-23
> > **Response to Reviewer WDbj**
> >
> > Dear Reviewer,
> >
> > We appreciate the opportunity to address your concerns and provide further clarifications. Below, we have outlined responses to your points:
> >
> > 1. As outlined in lines 189–193 of our revision, the embedding procedure indeed ensures graph isomorphism as long as the specified requirements are satisfied. Discussing purely topological equivalences (such as homeomorphism or homotopy equivalence) in this context is ambiguous, as the original graph does not inherently possess a natural topology.
> >
> > 2.
> >    - In the proof of Theorem 1, the symbol "$\approx$" denotes asymptotic equivalence. The phrase "since no sampling is involved" refers to our use of an equidistant partitioning of the relevant interval, avoiding any sampling approximation.
> >    - For Theorem 2, the statement "provides the necessary information for performing a single message-passing step" indicates that the respective l-$ECT_1$ allows us to reconstruct the feature vector information of all neighboring nodes. We regret any confusion caused by these points and will clarify them in our revised manuscript.
> >
> > 3. The fundamental insight here is that l-$ECT_1$ enables the recovery of feature vector information for all neighbors of the node in question. This property stems from the invertibility of l-$ECT$s, as utilized in Theorem 2. Importantly, l-$ECT$s offer a fixed-dimensional vector representation, even when nodes have varying numbers of neighbors. This capability is critical for using it as an expressive representation for downstream tasks.
> >
> > 4. We have substantially extended our experimental studies in the revised version to provide a more comprehensive evaluation.
> >
> > Please let us know if you have any additional questions or require further clarifications.
> >
> > If our responses adequately address your concerns, we would kindly request that you consider re-evaluating your overall rating.
> >
> > Best regards,
> > The Authors

---

> > > ### Author Response · Authors · 2024-11-27
> > >
> > > Dear Reviewer,
> > >
> > > In the latest revised version of our manuscript, we have addressed your concerns regarding our proofs. Additionally, we have included further explanations to emphasize that the expressivity of our method arises from its geometric-topological foundation, rather than being purely topological. Furthermore, we enhanced our numerical studies by conducting a post-hoc evaluation of our results, demonstrating both the general-purpose performance of our method and its out-of-the-box effectiveness on heterophilic graphs. The latter findings are detailed in the appendix. We are confident that we have adequately addressed all your concerns and would greatly appreciate it if you could kindly reconsider your overall evaluation.
> > >
> > > Best regards,
> > >
> > > the Authors

---

> > > ### Comment · Reviewer_WDbj · 2024-11-28
> > >
> > > The main issues remain. For example, the so-called embedding may encounter problems if there are drawn lines crossing each other in the Euclidean domain. The intuitions is that the proximity of the nodes in the graph and that according to the node features should be very different (particularly for heterophilic datasets). Therefore, the simple "embedding" described in the paper should not give a graph isomorphism.
> > >
> > > The revised proofs are still not acceptable. For example, "$\approx$ denotes asymptotic equivalence" does not mean anything. Instead of saying something such as $A_n\approx B_n$, an explicit upper bound of $||A_n-B_n||$ should be given. For another example, in Theorem 2, "one can reconstruct the feature vectors of its 1-hop neighborhood" remains imprecise. What does this mean exactly? Is there a precise reconstruction algorithm and is there an error bound? There are similar issues in other parts of the proofs.
> > >
> > > In my opinion, this paper contains technical flaws. In addition, my doubt on the usefulness of the added topological features for node classification remains. Hence, I do not recommend that the paper be accepted.

---

### Official Review · Reviewer_T69M · 2024-10-26

**Soundness:** 3
**Presentation:** 4
**Contribution:** 2
**Rating:** 6
**Confidence:** 5

**Summary:**

The authors introduce a new topological feature extraction methods, Local Euler Characteristic Transform (l-ECT), extending the Euler Characteristic Transform (ECT) to provide a lossless, interpretable representation of local graph neighborhoods, addressing limitations in traditional Graph Neural Networks (GNNs). This novel approach improves performance in node classification tasks, especially in heterophilous graphs, by preserving both local and global structural details.

**Strengths:**

1. **Novel l-ECT Framework**: Extending the Euler Characteristic Transform to capture local graph details in embedded simplicial complexes is impactful, with theoretical insights enhancing its expressivity, especially for featured graphs.

2. **Extracting  Key Information from Node Neighborhoods from Attribute Space**: The l-ECT enables to obtain node neighborhood information by effectively utilizing the information from attribute space.

3. **Experimental Validation**: The l-ECT consistently outperforms traditional GNNs in node classification tasks, particularly in high-heterophily settings, highlighting its interpretability and effectiveness.

4. **Presentation:** The presentation is very good.

**Weaknesses:**

1. **Limited Applicability:** The proposed approach is constrained to graphs with node feature vectors in $\mathbb{R}^n$, limiting its applicability to datasets that fit this specific structure.

2. **Effectiveness of Approach:** While the concept of embedding the graph into an attribute space using node attribute vectors is promising, the subsequent steps for extracting meaningful information appear less effective. The method could be enhanced by exploring simpler and more efficient ways to utilize the geometry (rather than topology) of ego networks induced within the attribute space.

3. **Feasibility in High Dimensions:** As the dimension $n$ of the feature space increases, the number $m$ of representative vectors on $S^{n-1}$ must grow nearly exponentially. Furthermore, the feature vector range impacts the number of intervals {$t_i$} needed. For high-dimensional and wide-range data, this results in a very high-dimensional $l$-ECT vector, making the approach impractical for real-world applications. Dimension reduction could help by reducing feature dimensionality to three (as two dimensions may be insufficient for graph embedding) and normalizing feature vectors (e.g. total diameter of feature vector space to 2), allowing for "end-to-end" a fixed-size feature extraction for nodes. Without this, selecting vectors and thresholds can be challenging, particularly for new users.

4. **Theoretical Contributions vs. Practical Applications:** While the rotation-invariant metric is mathematically appealing, it may lack practical relevance since it relies on the infimum over all rotations. Also, the discussion of graph isomorphism seems tangential, as Definition 2 is highly restrictive, applicable only to isomorphic graphs with identical feature vectors.

5. **Experimental Results:** The presented results are uninformative and potentially misleading. The models used, GCN and GAT, are older and are known to perform poorly in heterophilic settings. The authors should consider comparing their approach with newer GNN models that perform well on heterophilic datasets and include more homophilic datasets (other than Computers and Photo) to provide a comprehensive performance assessment. Also, exploring the integration of $l$-ECT vectors with a more recent GNN model may yield interesting insights into performance enhancement.

**Questions:**

See weaknesess.

---

> ### Author Response · Authors · 2024-11-20
> **Response to your review (1/2)**
>
> Dear Reviewer,
>
> thank you very much for carefully reviewing our paper. We are happy to hear that you acknowledge the novelty, utility and very good presentation of our results!
>
> Regarding your concerns:
>
> > Limited Applicability: The proposed approach is constrained to graphs with node feature vectors in Rn, limiting its applicability to datasets that fit this specific structure.
>
> Although this might look like a restriction at first, the assumption that node feature vectors lie in a (possibly high-dimensional) Euclidean space is a common one and applies to every node classification task the authors are aware of. Please let us know otherwise, we are happy to generalize our method accordingly!
>
> > Effectiveness of Approach: While the concept of embedding the graph into an attribute space using node attribute vectors is promising, the subsequent steps for extracting meaningful information appear less effective. The method could be enhanced by exploring simpler and more efficient ways to utilize the geometry (rather than topology) of ego networks induced within the attribute space.
>
> Although the approach of using local versions of Euler Characteristic Transforms is topological by nature, the resulting representation is in fact a lossless representation of the local graph neighborhood, containing both spatial information (of the respective feature vectors in the neighborhood) and structural information (of the graph structure of the neighborhood). Therefore, l-ECTs should not be seen as capturing pure topological information, but rather providing a fingerprint of ego graphs. We will clarify this aspect in our revision.
> However, we are happy to consider concrete suggestions which simplify our approach!
>
> > Feasibility in High Dimensions: As the dimension n of the feature space increases, the number m of representative vectors on Sn−1 must grow nearly exponentially. Furthermore, the feature vector range impacts the number of intervals {ti} needed. For high-dimensional and wide-range data, this results in a very high-dimensional l-ECT vector, making the approach impractical for real-world applications. Dimension reduction could help by reducing feature dimensionality to three (as two dimensions may be insufficient for graph embedding) and normalizing feature vectors (e.g. total diameter of feature vector space to 2), allowing for "end-to-end" a fixed-size feature extraction for nodes. Without this, selecting vectors and thresholds can be challenging, particularly for new users.
>
> You are correct in the sense that the number of representative vectors on the sphere grows exponentially in order to exceed a certain density threshold. However, it is known that only a fraction of representative directions suffices to obtain a lossless representation of the underlying graph (or simplicial complex) by using (l)-ECTs (see Justin Curry, Sayan Mukherjee, and Katharine Turner. How many directions determine a shape and other sufficiency results for two topological transforms. Transactions of the American Mathematical Society, Series B, 9(32):1006–1043, 2022.).
> The number of intervals is not affected by the range of the vectors since our l-ECT implementation ensures that we only start tracking information when the maximum (resp. minimum) value in the range is reached via a respective filtration parameter ti.
> In our experiments, we fixed both hyperparameters which determine the number of directions and interval steps to 60, which worked well even in situations with high-dimensional (several hundreds of dimensions) feature vectors. Moreover, an ablation study on the number of directions used is contained in the appendix and shows that often a very small number of directions suffices for reasonable results.

---

> > ### Author Response · Authors · 2024-11-20
> > **Response to your review (2/2)**
> >
> > > Theoretical Contributions vs. Practical Applications: While the rotation-invariant metric is mathematically appealing, it may lack practical relevance since it relies on the infimum over all rotations. Also, the discussion of graph isomorphism seems tangential, as Definition 2 is highly restrictive, applicable only to isomorphic graphs with identical feature vectors.
> >
> > Although the infimum for the rotation-invariant metric is taken over all directions, this leads to a well-defined learning procedure in practice (see Eq.9 in l.396), which yields reasonable results as can be seen from the experiments.
> > Definition 2 and the notion of subgraph counting stems from (Zhengdao Chen, Lei Chen, Soledad Villar, and Joan Bruna. Can graph neural networks count substructures? Advances in neural information processing systems, 33:10383–10395, 2020.)
> > We included this paragraph to show additional expressivity of our method, but it is not at the core of the paper.
> >
> > > Experimental Results: The presented results are uninformative and potentially misleading. The models used, GCN and GAT, are older and are known to perform poorly in heterophilic settings. The authors should consider comparing their approach with newer GNN models that perform well on heterophilic datasets and include more homophilic datasets (other than Computers and Photo) to provide a comprehensive performance assessment. Also, exploring the integration of l-ECT vectors with a more recent GNN model may yield interesting insights into performance enhancement.
> >
> > Thank you for highlighting this concern! Our method is designed as a general-purpose approach to graph learning, leveraging l-ECTs to move beyond the limitations of message-passing architectures. Consequently, we believe that comparing our method with general-purpose GNNs such as GCN and GAT, which are widely used baselines, provides meaningful insights into the versatility and applicability of our approach. This rationale aligns with our aim of emphasizing the general-purpose nature of our method rather than focusing on specialized architectures.
> > That said, we understand the interest in evaluating our method against models tailored for heterophilic settings. To address this, we will incorporate additional comparisons with a specialized heterophilic architecture such as H2GCN, as well as another general-purpose baseline like GIN, to provide a broader evaluation of our method's performance. These results will help contextualize the advantages of our approach while maintaining a balanced perspective.
> > Additionally, we agree that exploring the integration of l-ECTs with newer GNN models could provide valuable insights. While this is beyond the scope of the current paper, we will include a discussion in our revision outlining potential directions for such extensions, emphasizing how l-ECTs could complement modern GNN designs.
> > Finally, to address your concern about dataset diversity, we would like to highlight that results for the well-known Planetoid datasets are included in the appendix of our work. We are happy to include additional datasets.
> > In the meantime, please feel free to reach out if you have any additional questions. If our responses have sufficiently addressed your concerns, we would be grateful if you could consider re-evaluating your overall rating.
> >
> > Best regards,
> >
> > the Authors

---

> ### Comment · Reviewer_T69M · 2024-11-22
>
> 1. Thank you for clarifying the focus of your model. However, the concern about its applicability remains. Many benchmark datasets, such as citation networks, use binary node features https://ogb.stanford.edu/docs/nodeprop/. Since your model only works with continuous feature spaces in R^n, it is less useful for datasets with binary features. It would strengthen your work to address this limitation or explain how the model could be extended.
>
> 2. The embedding of ego networks in feature space is a promising approach for extracting meaningful node information. However, using the l-ECT on ego network embeddings may not provide the most useful features. The topology output often does not change with size or continuous shape changes, which might miss finer details. Geometric measures, such as the diameter or convex hull volume in R^n, could provide more meaningful information, especially with proper normalization in the feature space.
>
> 3.  I understand the time constraints during the rebuttal period. However, it was expected that you would include experiments with newer GNN models. These would show how your method compares to more recent approaches and make the results more convincing. Without these updates, this part of the work feels incomplete.
>
> Finally, as a reviewer, I find the use of exclamation marks in your responses inappropriate. I am providing feedback on your paper based on my expertise, and I would appreciate a more professional and respectful tone in your responses.

---

> > ### Author Response · Authors · 2024-11-22
> > **Response to Reviewer T69M**
> >
> > Dear Reviewer,
> >
> > We are sorry that our use of exclamation marks was perceived as inappropriate. We assure the reviewer that no disrespect from our side was intended. Given that this is textual communication, we wanted to communicate our enthusiasm and excitement. We shall refrain from doing this and apologise for this misunderstanding.
> >
> > Regarding your remaining concerns:
> >
> > 1. Binary features do not pose a restriction for our approach, provided that the requirements outlined in our construction are met. In fact, several datasets in our experiments include binary feature vectors and demonstrate reasonable results. This is because binary vectors can naturally be interpreted as residing in an R^n space, aligning with the assumptions of our model.
> > 2. We acknowledge your point about geometric measures potentially offering additional insights. However, due to the invertibility of the l-ECT, these geometric measures are inherently encoded within its output. The expressivity of our approach relies on this property, eliminating the need for explicit inclusion of such features. That said, we agree that other methodologies, such as those based on message-passing frameworks, could benefit from integrating these explicit geometric features.
> > 3. While our method is designed as a general-purpose approach and not specifically tailored for heterophilic graphs (as clarified in the paper), we appreciate the value of comparing our method against more recent architectures, including those designed for heterophily. We are happy to incorporate additional experiments with a heterophily-specific architecture to further strengthen our evaluation.
> >
> > We hope these clarifications address your concerns. If any questions remain, we would be glad to provide additional details.
> >
> > If our responses have sufficiently addressed your feedback, we would be grateful if you could kindly consider re-evaluating your overall assessment.
> >
> > Thank you once again for your thoughtful review and constructive suggestions.
> >
> >
> > Best regards,
> >
> > the Authors

---

> > > ### Author Response · Authors · 2024-11-27
> > >
> > > Dear Reviewer,
> > >
> > > We want to mention again that no disrespect was intended in our previous messages. We hope to have shown that we take your feedback seriously and appreciate your help in improving our work. In the meantime, we have revised our manuscript to include additional clarifications on our method, providing an intuitive explanation of why our approach is expressive. Furthermore, we would like to reiterate that, although the node classification experiments (except for the two datasets Roman Empire and Amazon Ratings) in our main paper involve binary feature vectors, our method demonstrates strong performance. To address your concern regarding comparisons with additional GNN models, we have included a post-hoc evaluation in the appendix. We now believe that we have thoroughly addressed all your concerns and would greatly appreciate it if you could kindly consider re-evaluating your overall score.
> > >
> > > Best regards,
> > >
> > > the Authors

---

> > > > ### Comment · Reviewer_T69M · 2024-11-28
> > > >
> > > > Thanks for your responses. Most of my concerns are addressed. I am raising my score.

---

> > > > > ### Author Response · Authors · 2024-11-28
> > > > >
> > > > > Dear Reviewer,
> > > > >
> > > > > Thank you for your detailed feedback throughout the review process. We greatly appreciate the time and effort you dedicated to providing constructive comments and suggestions to help improve our work.
> > > > > We are especially grateful that our revisions and clarifications addressed most of your concerns and that you were willing to reconsider your score.
> > > > > Thank you once again for your support and for contributing to the development of our work.
> > > > >
> > > > > Best regards,
> > > > >
> > > > > the Authors

---

### Official Review · Reviewer_pj4s · 2024-11-03

**Soundness:** 3
**Presentation:** 3
**Contribution:** 3
**Rating:** 6
**Confidence:** 4

**Summary:**

This paper introduces the Local Euler Characteristic Transform ($l$-ECT), an extension of the Euler Characteristic Transform (ECT) designed to enhance expressivity and interpretability in graph representation learning. It provides a lossless representation of local neighborhoods and addresses key limitations in GNNs by preserving nuanced local structures while maintaining global interpretability. Their method demonstrates superior performance over standard GNNs on node classification tasks, particularly in graphs with heterophily.

**Strengths:**

1. Innovative Use of Euler Characteristic Transform: Employing the ECT to enhance graph representation learning, especially in settings with heterophily, is a novel and interesting approach.

2. Solid Theoretical Foundation: The work is thorough, with strong theoretical results that effectively support the proposed method.

**Weaknesses:**

Missing Important Related Works & Limited Experimental Comparisons: The quantitative experiments focus mainly on node classification tasks in heterophilic graphs but compare the proposed method only with basic models like GCN and GAT. While the authors acknowledge related works on GNNs designed for heterophily in Section 3, the coverage is still limited. It is suggested that the authors include more related works such as [1-5] and select appropriate GNNs for experimental comparison to strengthen the validation of their method.

[1] Beyond Homophily in Graph Neural Networks: Current Limitations and Effective Designs

[2] Graph Neural Networks with Heterophily

[3] Predicting Global Label Relationship Matrix for Graph Neural Networks under Heterophily

[4] ES-GNN: Generalizing Graph Neural Networks Beyond Homophily With Edge Splitting

[5] GBK-GNN: Gated Bi-Kernel Graph Neural Networks for Modeling Both Homophily and Heterophily

**Questions:**

See weaknesses.

---

> ### Author Response · Authors · 2024-11-20
> **Response to your review**
>
> Dear Reviewer,
>
> Thank you for your feedback! Our method is designed as a general-purpose approach to graph learning that inherently works well for heterophilic graphs due to its mechanistic design. Specifically, by leveraging l-ECTs, we introduce an alternative paradigm that moves beyond the fundamental limitations of message-passing approaches. Unlike specialized architectures tailored for heterophilic graphs, our method does not rely on task-specific adaptations or additional mechanisms, which underscores its versatility and applicability across various graph settings.
>
> Given this general-purpose nature, we believe that the most meaningful comparisons are with other general-purpose GNNs, such as GCN or GAT, which are not specifically designed for heterophily but are widely used as baselines across a range of tasks.
> While we understand the interest in comparing against architectures designed specifically for heterophilic graphs, we argue that such comparisons might not be entirely fair, as these models include domain-specific mechanisms that directly target heterophily. By contrast, our method's strength lies in its ability to perform well on heterophilic graphs without such explicit tailoring. Nevertheless, in the spirit of thoroughness and to address your concern, we are happy to include results comparing our method against one specialized architecture like H2GCN as suggested by you, in addition to the general-purpose GNNs. Moreover, we will add results for one other general-purpose architecture, such as GIN.
>
> Finally, we will incorporate a discussion on how our findings relate to the insights provided in "A critical look at the evaluation of GNNs under heterophily: Are we really making progress?" to situate our contributions within the broader discourse on heterophilic graph learning. We hope this approach demonstrates the unique advantages of our method and emphasizes its general-purpose design while maintaining a balanced perspective on its evaluation.
>
> In the meantime, please feel free to reach out if you have any additional questions. If our responses have sufficiently addressed your concerns, we would be grateful if you could consider re-evaluating your overall rating.
> Once again, we thank you for the support of our work!
>
> Best regards,
>
> the Authors

---

> > ### Comment · Reviewer_pj4s · 2024-11-24
> >
> > Thanks for your reponse and I'd like to keep my score.

---

### Official Review · Reviewer_Mpt1 · 2024-11-04

**Soundness:** 3
**Presentation:** 3
**Contribution:** 3
**Rating:** 8
**Confidence:** 3

**Summary:**

The paper introduces the Local Euler Characteristic Transform (L-ECT), an extension of the Euler Characteristic Transform (ECT) designed for graph representation learning. Unlike traditional Graph Neural Networks (GNNs), which can obscure local details through node aggregation, the L-ECT maintains local structural data, thus enhancing interpretability and performance, especially in heterogeneous (high heterophily) graphs. By capturing spatial and structural characteristics of local neighborhoods, the L-ECT provides a rotation-invariant metric for data alignment, showcasing improved performance over GNNs in node classification tasks. The method’s compatibility with machine learning models enables use cases beyond standard GNN architectures, offering more accessible and interpretable models, such as tree-based classifiers. Empirical results demonstrate that L-ECT outperforms GNNs in heterogeneous datasets and facilitates robust spatial alignment in both synthetic and high-dimensional data. This research suggests future exploration into scaling L-ECT and integrating global and local information in complex graph structures.

**Strengths:**

The paper presents the Local Euler Characteristic Transform (L-ECT) as an extension of the traditional Euler Characteristic Transform, enabling a lossless representation of local graph structures and addressing key limitations of Graph Neural Networks (GNNs) such as oversmoothing and loss of local detail in high heterophily graphs. This novel transformation preserves intricate topological information, allowing for more nuanced node representations by capturing both structural and spatial data and offering an alternative to GNN message-passing frameworks. Additionally, the authors introduce a rotation-invariant metric that enables robust spatial alignment of data in Euclidean space, enhancing the method’s applicability in graph-structured data and increasing resilience to coordinate transformations. Empirical results underscore L-ECT’s effectiveness, showing superior performance over standard GNNs in high-heterophily datasets like WebKB, Roman Empire, and Amazon Ratings. Furthermore, L-ECT’s model-agnostic nature facilitates integration with interpretable machine learning models, such as XGBoost, making it ideal for use in regulated fields like healthcare and finance where transparency is paramount. Beyond graph representation, L-ECT extends to point clouds and other high-dimensional data, proving robust to noise and outliers and enabling efficient spatial alignment without the need for exhaustive pairwise distance computations.

The methods section is detailed yet readable, presenting L-ECT’s mathematical foundation and integrating a rotation-invariant metric for spatial alignment, which adds to the paper’s originality. While the experiments section is robust and results are well-presented through tables and figures, additional visual aids could further clarify data characteristics and enhance accessibility.

the discussion on the limitations of the approaches proposed in the paper is appreciated

**Weaknesses:**

The paper would benefit, both in making more persuasive the novelty of the work with respect to contemporary literature as well as clarity of the work itself, with a more robust background and related works section

Including a more robust and explicit comparison to related works, which also addresses the novelty of the work being proposed, would be appreciated.

The L-ECT approach, while innovative, faces several limitations and lacks certain aspects of novelty. Its computational complexity scales with graph size and density, making it less efficient for very large or dense graphs and primarily feasible for medium-sized datasets. Although L-ECT emphasizes local information preservation, similar topology-aware or geometric GNN approaches also capture neighborhood-specific details, reducing the uniqueness of this feature. Additionally, traditional GNNs perform comparably well on low-heterophily datasets, indicating that L-ECT may not consistently outperform them across all types of graph data. The approach’s scalability is further limited by sampling trade-offs, as its accuracy depends on carefully chosen parameters, such as direction and filtration steps, which challenge fidelity and computational efficiency at scale. Moreover, despite its model-agnostic design, L-ECT’s interpretability hinges on pre-defined features, potentially restricting its flexibility for complex, dynamic graphs. Finally, L-ECT does not support end-to-end learning as GNNs do; instead, it relies on external classifiers (e.g., XGBoost), which may limit its integration into more comprehensive, end-to-end pipelines.

The authors should include comparison other works which construct topological representations of graphs and graphs neighborhoods and include reference to those related methods such as “graph filtration learning” by Hofer et. al. and other approaches as discussed in survey works such as  “A Survey of Topological Machine Learning Methods” by Hensel et. al.

The authors provide comparative experimental analysis to a number of datasets. It may be misleading, however, to not include other models as discussed in “A critical look at the evaluation of GNNs under heterophily: Are we really making progress?” by Platonov et. al.

**Questions:**

Would it be possible to include experimental results for the other datasets offered in “A critical look at the evaluation of GNNs under heterophily: Are we really making progress?” by Platonov et. al. or an argument as to why this is done?

---

> ### Author Response · Authors · 2024-11-20
> **Response to your review (1/2)**
>
> Dear Reviewer,
>
> thank you for your thoughtful review and for recognizing the strengths of our paper, particularly the l-ECT’s capacity to enhance interpretability and performance in heterophilic graphs and its model-agnostic design.
>
> Regarding your concerns:
>
> > The paper would benefit, both in making more persuasive the novelty of the work with respect to contemporary literature as well as clarity of the work itself, with a more robust background and related works section
>
> Thank you for this suggestion! We will extend the Background and Related Work sections, and sharpen the novelty of our work, in our revision. To our knowledge, there is indeed no other work making use of local variants of the ECT at this point.
>
> > The L-ECT approach, while innovative, faces several limitations and lacks certain aspects of novelty. Its computational complexity scales with graph size and density, making it less efficient for very large or dense graphs and primarily feasible for medium-sized datasets.
>
> We are aware of this limitation (see l.234-240) and work on extensions of the proposed method for future work, so that it is also applicable for large and dense graphs. We will clarify this fact better in a revision.
>
> > Although L-ECT emphasizes local information preservation, similar topology-aware or geometric GNN approaches also capture neighborhood-specific details, reducing the uniqueness of this feature.
>
> To the best of our knowledge, such topology-aware and geometric GNN approaches are usually based on message passing. Our approach overcomes the fundamental limitation induced by message passing, introducing a novel and interpretable paradigm that allows for graph neighborhood representation without the need of aggregating neighboring feature vector information. Given the recent insights into the fundamental limitations of architectures based on message passing (https://arxiv.org/abs/2408.05486), we believe that our work outlines new research avenues to pursue.
>
> > Additionally, traditional GNNs perform comparably well on low-heterophily datasets, indicating that L-ECT may not consistently outperform them across all types of graph data.
>
> Indeed, we do not claim to have found a new state-of-the art method for benchmarking graph datasets (see l.259-263), but we rather propose a fundamentally different approach for graph learning that overcomes fundamental limitations introduced by message passing. As the main advantage of our method, we see its model-agnosticism, interpretability and the different mechanistic design which does not necessitate on node feature vector aggregation.
>
> > Moreover, despite its model-agnostic design, L-ECT’s interpretability hinges on pre-defined features, potentially restricting its flexibility for complex, dynamic graphs. Finally, L-ECT does not support end-to-end learning as GNNs do; instead, it relies on external classifiers (e.g., XGBoost), which may limit its integration into more comprehensive, end-to-end pipelines.
>
> Making our method end-to-end learnable is also a direction which we leave for future work. However, we do not see any fundamental obstruction in doing so since the model agnosticism of the method allows for using neural networks for classification. However, the focus of this work was to introduce an approach to graph learning which is both interpretable and applicable to data-scarce scenarios.

---

> > ### Author Response · Authors · 2024-11-20
> > **Response to your review (2/2)**
> >
> > > The authors should include comparison other works which construct topological representations of graphs and graphs neighborhoods and include reference to those related methods such as “graph filtration learning” by Hofer et. al. and other approaches as discussed in survey works such as “A Survey of Topological Machine Learning Methods” by Hensel et. al.
> >
> > Thank you very much for this suggestion! We will include these into our Related Work section. We will also provide a comparison to such methods, noting that they are often, unfortunately, not capable of node classification.
> >
> > > The authors provide comparative experimental analysis to a number of datasets. It may be misleading, however, to not include other models as discussed in “A critical look at the evaluation of GNNs under heterophily: Are we really making progress?” by Platonov et. al.
> >
> > We see our method as the first generic attempt to use l-ECTs for graph learning, introducing an alternative paradigm in order to overcome fundamental limitations incorporated by message passing. Other models in the reference you mention use specialized architectures with mechanisms that are not used in our approach due to its genericity. We therefore propose to compare our method against general purpose GNNs, such as GAT and GCN. However, we are happy to include comparisons against one specialized message-passing based architecture such as H2GCN, and against one other generic GNN, such as GIN, in our revision. Moreover, we will add a discussion on how our results relate to the findings given in  “A critical look at the evaluation of GNNs under heterophily: Are we really making progress?”
> > Regarding the other datasets, the size and density of the respective graphs posed challenges that prevented us from obtaining results. Addressing the scalability of the proposed method is left for future work, and we welcome suggestions on this topic!
> >
> >
> > In the meantime, please feel free to reach out if you have any additional questions. Once again, we thank you for the support of our work!
> >
> > Best regards,
> >
> > the Authors

---

> > > ### Comment · Reviewer_Mpt1 · 2024-11-26
> > >
> > > I appreciate the authors thorough response, addressing all points raised, as well as incorporating questions regarding related work and background. I believe the authors to have a strong paper.

---

### Author Response · Authors · 2024-11-20

Dear Reviewers and Area Chair,

We sincerely thank you for your thoughtful reviews and constructive feedback.
We are currently running experiments with the goal of further improving our results,
and working on our revision to address your concerns.

In the meantime, please let us know if there are any further questions!

Best regards,

the Authors

---

### Author Response · Authors · 2024-11-22
**Comment on Revision**

Dear Reviewers,

We thank you for your valuable feedback and detailed assessments of our submission. We are pleased to inform you that we have carefully addressed the points raised and uploaded a revised version of the manuscript, with changes highlighted in green. Below, we summarize the updates made in the revision:

We have expanded the Background and Related Work sections to include additional references and provide a more robust comparison with existing methods, as requested. The experimental comparisons have been extended to include results for the additional specialized model H2GCN and the general-purpose baseline GIN. We have also discussed the implications of the results in the context of related literature. The embedding process has been clarified, including the distinction between graph and simplicial complex embeddings. Scalability and dimensionality concerns have been addressed by elaborating on strategies to tackle these challenges, including potential future extensions to support larger and denser graphs.

We believe these revisions significantly strengthen the submission by addressing the concerns raised, providing additional insights, and clarifying key aspects of our approach.

If our changes satisfactorily address your concerns, we kindly ask you to consider adapting your overall rating of your review. Your constructive feedback has been instrumental in improving the quality and rigor of our work, and we are grateful for your time and effort.

Best regards,
The Authors

---

### Author Response · Authors · 2024-11-27
**Comment on latest revision**

Dear Reviewers and Area Chair,

We sincerely appreciate the reviewers' valuable feedback, which has provided us with meaningful opportunities to improve the manuscript. Owing to the unavailability of a more stringent comparison of general-purpose methods in the literature, next to the new results we included in the revision, we will also work on integrating more methods into our experimental setup, facilitating a comprehensive comparison. We believe this to be a substantial endeavour and are confident to have addressed the reviewers' concerns.

Best regards,

the Authors

---

### Meta-Review · Area_Chair_j1KT · 2024-12-19

**Metareview:**

The paper introduces the Local Euler Characteristic Transform (l-ECT), an extension of the Euler Characteristic Transform (ECT) aimed at enhancing expressivity and interpretability in graph representation learning. Unlike traditional GNNs, which may lose important local details during aggregation, l-ECT provides a lossless representation of local neighborhoods, preserving nuanced structures while maintaining global interpretability. Additionally, the authors propose a rotation-invariant metric based on l-ECT for spatial alignment of data. Experimental results show that l-ECT outperforms standard GNNs, particularly in high-heterophily node classification tasks.

### Strengths:

1. The use of the ECT to enhance graph representation learning, especially in high-heterophily settings, is both novel and compelling. The authors provide solid theoretical backing to support the proposed method.

2. The l-ECT consistently outperforms traditional GNNs in node classification tasks, particularly in high-heterophily environments, demonstrating its interpretability and effectiveness.

### Weaknesses:

1. The technical sections, including the proofs, are poorly presented, making it difficult to assess the correctness of the results.

2. The numerical results are not particularly striking, and the experimental design could be strengthened to more convincingly demonstrate that the approach advances the state of the art.

### Overall:

This paper presents an interesting and novel idea by applying l-ECT to graph learning. However, the clarity of the technical sections is insufficient, which hampers the ability to evaluate the correctness of the method. Additionally, the experimental results are not compelling enough to clearly show the method’s superiority. As a result, I recommend borderline rejection, but strongly encourage the authors to address the reviewers' suggestions and revise the paper accordingly.

**Additional Comments On Reviewer Discussion:**

During the rebuttal period, the authors addressed the following points:

- In response to Reviewer Mpt1, pj4s, and JKXe, the authors provided additional explanations and results to address concerns regarding comparisons with contemporary literature, computational complexity, basic models, and experimental results. These three reviewers have generally expressed satisfaction with the authors' clarifications and efforts.

- Reviewer WDbj raised significant concerns about the correctness of the Euclidean embedding. In response, the authors explained that the embedding procedure ensures graph isomorphism, provided the specified requirements are met. Although the reviewer insisted that the map to the Euclidean space should either be a homomorphism (onto the image) or a diffeomorphism, depending on the required properties, I believe these conditions are not directly relevant to the approach presented. Nevertheless, I agree that the writing should be improved, and more details should be added, including background on simplicial complexes, the rationale behind the Euclidean embedding, and the associated proofs.

---

### Decision · Program_Chairs · 2025-01-22

Reject